# Building Math Agents with Multi-Turn Iterative Preference Learning

**Wei Xiong** [1], **Chengshuai Shi**[2], **Jiaming Shen**[3], **Aviv Rosenberg**[4], **Zhen Qin**[3], **Daniele Calandriello**[3]
**Misha Khalman**[3], **Rishabh Joshi**[3], **Bilal Piot**[3], **Mohammad Saleh**[3], **Chi Jin**[5], **Tong Zhang**[1], **Tianqi Liu**[3]
University of Illinois Urbana-Champaign[1]
University of Virginia[2]
Google Deepmind[3]
Google Research[4]
Princeton University[5]
{wx13, tozhang}@illinois.edu; tianqiliu@google.com

## Abstract

Recent studies have shown that large language models' (LLMs) mathematical problem-solving capabilities can be enhanced by integrating external tools, such as code interpreters, and employing multi-turn Chain-of-Thought (CoT) reasoning. While current methods focus on synthetic data generation and Supervised Fine-Tuning (SFT), this paper studies the complementary direct preference learning approach to further improve model performance. However, existing direct preference learning algorithms are originally designed for the single-turn chat task, and do not fully address the complexities of multi-turn reasoning and external tool integration required for tool-integrated mathematical reasoning tasks. To fill in this gap, we introduce a multi-turn direct preference learning framework, tailored for this context, that leverages feedback from code interpreters and optimizes trajectory-level preferences. This framework includes multi-turn DPO and multi-turn KTO as specific implementations. The effectiveness of our framework is validated through training of various language models using an augmented prompt set from the GSM8K and MATH datasets. Our results demonstrate substantial improvements: a supervised fine-tuned Gemma-1.1-it-7B model's performance increased from 77.5% to 83.9% on GSM8K and from 46.1% to 51.2% on MATH. Similarly, a Gemma-2-it-9B model improved from 84.1% to 86.3% on GSM8K and from 51.0% to 54.5% on MATH.

## 1 Introduction

Large language models (LLMs) have demonstrated remarkable capacities across a variety of language tasks. Notable models include ChatGPT (OpenAI, 2023), Claude (Anthropic, 2023), and Gemini (Gemini et al., 2023). However, despite these advances, even the most advanced closed-source LLMs still struggle with complex reasoning tasks that require multi-turn decision making. In particular, for the representative task of mathematical problem solving, LLMs often fail with basic arithmetic and symbolic computations (Hendrycks et al., 2021; Zheng et al., 2021). To address this issue, recent studies recommend the integration of external tools (e.g., calculators, computational Python libraries and symbolic solvers) to augment the LLMs' mathematical problem-solving capabilities (Shao et al., 2022; Mishra et al., 2022; Zhang et al., 2024a). Specifically, by integrating natural language reasoning with the use of these external tools, these enhanced LLMs can receive external messages from tool interactions and reason based on both previously generated tokens and external messages, which significantly improves their performance in mathematical tasks (Gou et al., 2023b; Toshniwal et al., 2024; Shao et al., 2024).

These successes of tool-integrated LLMs lead to a natural research question: how can we better train LLMs to combine tool usage with intrinsic reasoning to tackle complex reasoning tasks? For mathematical problem solving, existing works primarily focus on synthetic data generation (by strong teacher models) and supervised fine-tuning (SFT), as seen in ToRA (Gou et al., 2023b), Meta-MathQA (Yu et al., 2023), MAmmoTH (Yue et al., 2023; 2024), and Open-MathInstruct (Toshniwal

et al., 2024). These synthetic datasets have yielded significant improvements in test accuracy on standard benchmarks like MATH (Hendrycks et al., 2021) and GSM8K (Cobbe et al., 2021a).

Built on strong SFT models, *Reinforcement Learning from Human Feedback* (RLHF) has proven to be a key technique to elicit LLMs' knowledge during the post-training stage and has become standard in the LLM training pipeline (Ouyang et al., 2022; Gemini et al., 2023). Broadly speaking, the RLHF learning paradigm, which was originally designed for aligning LLMs with human values and preferences, is distinct from SFT as it learns from *relative feedback*. It has notably enhanced the capabilities of models like ChatGPT, Claude, and Gemini, enabling them to generate responses that are more helpful, harmless, and honest (Bai et al., 2022). Inspired by RLHF's success in general chat applications, in this paper, we explore RLHF for improving LLMs' mathematical problem-solving abilities when equipped with external tools. In particular, since deep RL methods (e.g., the proximal policy optimization, PPO algorithm (Schulman et al., 2017)) are often sample inefficient and unstable (Choshen et al., 2019), our goal is to derive direct preference learning algorithms that directly learn from the preference dataset (Zhao et al., 2023; Rafailov et al., 2023).

**Contribution.** We begin by formulating the learning process as a Markov decision process (MDP), distinct from the contextual bandit approach typically used in RLHF for making general chatbots without external environment interactions (Xiong et al., 2024; Rafailov et al., 2023). Then, we derive the optimality condition of the planning with such an MDP and our findings indicate that when the external randomness is low, we can develop multi-turn direct alignment algorithms (M-DPO and M-KTO), where the primary modification is to mask out irrelevant tokens during training. Furthermore, we extend our approach to its online iterative variants, which recent works demonstrated to be promising (Xiong et al., 2024; Guo et al., 2024b). Finally, we evaluate our approach through case studies using augmented training sets from MATH and GSM8K benchmarks, employing various base models such as Gemma (Team et al., 2024), CodeGemma (Team, 2024), and Mistral (Jiang et al., 2023). For instance, the performance of a supervised fine-tuned Gemma-1.1-it-7B model increased from 77.5% to 83.9% on GSM8K and from 46.1% to 51.2% on MATH. Similarly, a Gemma-2-it-9B model improved from 84.1% to 86.3% on GSM8K and from 51.0% to 54.5% on MATH. These empirical results indicate a significant improvement in performance over standard SFT models, demonstrating the potential of RLHF in complex reasoning task. We also provide a comprehensive recipe for the practical implementation of our online iterative multi-turn methods, and will make our models, datasets, and code publicly available for further research and development.

## 2 ALGORITHMS DEVELOPMENT

### 2.1 PROBLEM FORMULATION

We first formally formulate the tool-integrated reasoning task. At the first step, a prompt $x \in \mathcal{X}$ is sampled from some distribution $d_0$ as the initial state $s_1 = x$. Then, at each step $h \in [H]$,

- **Action:** the agent observes the current state $s_h$, which is the history of the first $h - 1$ interactions with the external environment, and takes an action $a_h$ according to some policy $\pi_h(\cdot|s_h) \in \Delta(\mathcal{A})$.

- **Observation:** in response to the agent's action, the environment then returns an observation $o_h \sim \mathbb{P}_h^*(\cdot|s_h, a_h)$[1] based on the history $s_h$ and current action $a_h$.

Then, we transit to a new state, which is the history up to the step $h + 1$: $s_{h+1} = (s_h, a_h, o_h) = (x, a_1, o_1, \cdots, a_h, o_h)$, and a new step begins. This process repeats for $H$ rounds in total and eventually, we collect a trajectory: $\tau = (x, a_1, o_1, \cdots, o_{H-1}, a_H)$. We present an example of multi-turn tool-integrated reasoning in Figure 4. Typically, the action is in the ReAct manner, which consist of a reasoning step $f_h$ and an execution step $e_h$ (e.g., writing python code) (Yao et al., 2022). We mention in passing that such an MDP formulation of preference learning was recently studied in Zhong et al. (2024); Rafailov et al. (2024); Xie et al. (2024a) but with a focus on the single-turn chat task and without explicitly considering the external messages.

---

[1] When there is no ambiguity, the abbreviation $s_{h+1} \sim \mathbb{P}_h^*(\cdot|s_h, a_h)$ is also adopted.

To connect the problem with RLHF that learns from *relative feedback*, we follow Ouyang et al. (2022); Bai et al. (2022) to assume that we can query the Bradley-Terry model for preference signal.

**Definition 1** (Bradley-Terry model). *We denote $\tau/x = y$, where the prompt is excluded from the trajectory. We assume that there exists a utility function of the trajectory $u^*$ such that given $(x, y^1, y^2)$, one response $y^1$ is preferred over another response $y^2$, denoted as $y^1 \succ y^2$, with probability*

$$\text{Prob}\big(y^1 \succ y^2 \mid x, y^1, y^2\big) = \sigma\big(u^*(x, y^1) - u^*(x, y^2)\big), \tag{1}$$

*where $\sigma$ is the sigmoid function $\sigma(z) = 1/(1 + \exp(-z))$. Also, given $(x, y^1, y^2)$ we denote the sampled preference signal as $z$ with $z = 1$ indicating $y^1 \succ y^2$ while $z = 0$ indicating $y^2 \succ y^1$.*

Here we only assume access to the trajectory-level preference, but not an action-level one. However, we remark that the utility function itself can be defined in a step-wise manner. Examples of the utility function include the binary reward from checking final result, outcome-supervised reward models (Cobbe et al., 2021b), and process-supervised reward model (Lightman et al., 2023).

## 2.2 Planning with a Model: Optimality Condition and practical Algorithm

We develop the main algorithms in this section with the general MDP formulation. Following Rafailov et al. (2023), we first establish the connection between a model $\mathcal{M} = (\mathcal{S}, \mathcal{A}, H, \mathbb{P}, d_0, u)$ and its associated optimal policy. In particular, we are interested in the following KL-regularized planning problem with respect to a reference policy $\pi_{\text{ref}}$:

$$\arg\max_{\pi} J(\pi; \mathcal{M}, \pi_{\text{ref}}) = \mathbb{E}_{x \sim d_0, a_h \sim \pi_h(\cdot|s_h), o_h \sim \mathbb{P}_h(\cdot|s_h, a_h)} \Big[ u(x, y) - \eta \sum_{h=1}^{H} D_{\text{KL}}\big(\pi_h(\cdot|s_h), \pi_{\text{ref},h}(\cdot|s_h)\big) \Big]. \tag{2}$$

In the single-turn case with $H = 1$, the optimal solution with respect to a utility function $u$ is the *Gibbs distribution* (see Lemma 3). Moving toward multi-turn case, we first consider $H = 2$ to illustrate the idea. The idea is to take a backward iteration from $h = H = 2$ to $h = 1$. Specifically, when we fix $s_2$ and consider only the step 2, it reduces to the single-turn case:

$$\pi_{\mathcal{M},2}(\cdot|s_2) = \arg\max_{\pi_2} \mathbb{E}_{a_2 \sim \pi_2(\cdot|s_2)} \Big( u(s_2, a_2) - \eta \cdot D_{\text{KL}}\big(\pi_2(\cdot|s_2), \pi_{\text{ref},2}(\cdot|s_2)\big) \Big) \propto \pi_{\text{ref},2}(\cdot|s_2) \cdot \exp\Big(\frac{u(s_2, \cdot)}{\eta}\Big).$$

Then, we can define the value function associated with $\pi_{\mathcal{M},2}$ as

$$V_{\mathcal{M},2}(s_2) := \mathbb{E}_{a_2 \sim \pi_{\mathcal{M},2}(\cdot|s_2)} \big[ u(s_2, a_2) - \eta D_{\text{KL}}\big(\pi_{\mathcal{M},2}(\cdot|s_2), \pi_{\text{ref},2}(\cdot|s_2)\big) \big]$$
$$Q_{\mathcal{M},1}(s_1, a_1) := \mathbb{E}_{o_1 \sim \mathbb{P}_1(\cdot|s_1, a_1)} \big[ V_{\mathcal{M},2}(s_2) \big].$$

For step 1, since we have determined $\pi_{\mathcal{M},2}$, with the definition of $Q_{\mathcal{M},1}(s_1, a_1)$, we have

$$\pi_{\mathcal{M},1}(\cdot|s_1) = \arg\max_{\pi_1} \mathbb{E}_{a_1 \sim \pi_1(\cdot|x)} \Big[ Q_{\mathcal{M},1}(s_1, a_1) - \eta D_{\text{KL}}\big(\pi_1(\cdot|s_1), \pi_{\text{ref},1}(\cdot|s_1)\big) \Big] \propto \pi_{\text{ref},1}(\cdot|s_1) \cdot \exp\Big(\frac{Q_{\mathcal{M},1}(s_1, \cdot)}{\eta}\Big).$$

By construction, $\{\pi_{\mathcal{M},h}\}_{h=1}^{2}$ is optimal as it maximizes the KL-regularized target. For general MDP, we can repeat the process for $H$ times starting with $V_{\mathcal{M},H+1} = 0$ where we recursively define

$$Q_{\mathcal{M},h}(s_h, a_h) = \begin{cases} u(s_H, a_H), & \text{if } h = H, \\ \mathbb{E}_{o_h \sim \mathbb{P}_h(\cdot|s_h, a_h)}[V_{\mathcal{M},h+1}(s_{h+1})], & \text{if } h \leq H - 1, \end{cases} \tag{3}$$

Here the optimal policy and the $V$-values are given by

$$\pi_{\mathcal{M},h}(a_h|s_h) := \frac{1}{Z_h(s_h)} \pi_{\text{ref},h}(a_h|s_h) \cdot \exp\Big(\frac{Q_{\mathcal{M},h}(s_h, a_h)}{\eta}\Big) \qquad \text{(Gibbs distribution of } Q_{\mathcal{M},h})$$

$$V_{\mathcal{M},h}(s_h) := \mathbb{E}_{a_h \sim \pi_{\mathcal{M},h}(\cdot|s_h)} \big[ Q_{\mathcal{M},h}(s_h, a_h) - \eta \cdot D_{\text{KL}}\big(\pi_{\mathcal{M},h}(\cdot|s_h), \pi_{\text{ref},h}(\cdot|s_h)\big) \big], \tag{4}$$

$$= \eta \log \mathbb{E}_{\pi_{\text{ref},h}(a_h'|s_h)} \exp\Big(\frac{Q_{\mathcal{M},h}(s_h, a_h')}{\eta}\Big) = \eta \log Z_h(s_h),$$

where $Z_h(s_h) = \sum_{a_h \in \mathcal{A}} \pi_{\text{ref},h}(a_h|s_h) \cdot \exp\big(\frac{Q_{\mathcal{M},h}(s_h, a_h)}{\eta}\big)$ is the normalization constant. The second equality in the definition of the $V$-value is from Lemma 3. Then, by definition, $[\pi_{\mathcal{M},h}]_{h=1}^{H}$ is optimal. Essentially, we solve $H$ Gibbs distributions in terms of the $Q$-values. We summarize the results into the following proposition.

**Proposition 1.** *We can recursively define the following optimal value functions and optimal policies for a KL-regularized MDP with horizon $H$ and external observation $o_h$. For Q value, we have*

$$Q_{\mathcal{M},h}(s_h, a_h) = \begin{cases} u(s_H, a_H), & \text{if } h = H, \\ \mathbb{E}_{o_h \sim \mathbb{P}_h(\cdot|s_h, a_h)}[V_{\mathcal{M},h+1}(s_{h+1})], & \text{if } h \leq H - 1. \end{cases} \tag{5}$$

*Also, for all $h \in [H]$, we have:*

$$V_{\mathcal{M},h}(s_h) = \eta \log \underbrace{\mathbb{E}_{a_h \sim \pi_{\mathrm{ref},h}(\cdot|s_h)} \exp\left(\frac{Q_{\mathcal{M},h}(s_h, a_h)}{\eta}\right)}_{=:Z_h(s_h)},$$

$$\pi_{\mathcal{M},h}(a_h \mid s_h) = \frac{\pi_{\mathrm{ref},h}(a_h \mid s_h)}{Z_h(s_h)} \cdot \exp\left(\frac{Q_{\mathcal{M},h}(s_h, a_h)}{\eta}\right). \tag{6}$$

We have a few interesting observations that may be of independent interests.

1. The optimal value function is characterized by the expectation with respect to the initial reference policy due to the additional KL constraint.

2. For a fixed step $h$ and state-action pair $(s_h, a_h)$, we can treat the future as a bandit (with only one step), then, we have $Q_{\mathcal{M},h}(s_h, a_h) = \mathbb{E}_z u(s_h, a_h, z)$, where $z$ is a completion staring from $(s_h, a_h)$. One can use the Monte-Carlo estimation to estimate this value by multiple roll-outs. We notice that the non-regularized version of this process, is commonly referred to as the *process-supervised reward* (PRM) in the literature (Wang et al., 2023a). In other words, the PRM constructed in Wang et al. (2023a) is essentially a Q learning process.

We remark that the results are essentially from the entropy-regularized MDPs (Williams & Peng, 1991; Ziebart, 2010).

**Multi-turn DPO.** According to equation 4, we can solve the $Q$-values as

$$Q_{\mathcal{M},h}(s_h, a_h) = \eta \cdot \log \frac{\pi_{\mathcal{M},h}(a_h|s_h)}{\pi_{\mathrm{ref},h}(a_h|s_h)} + V_{\mathcal{M},h}(s_h). \tag{7}$$

Furthermore, combining equation 7 with the definition of $Q$-values $Q_{\mathcal{M},h}$ in equation 3, we have

$$\mathbb{E}_{o_h \sim \mathbb{P}_h(\cdot|s_h, a_h)} V_{\mathcal{M},h+1}(s_{h+1}) = \eta \cdot \log \frac{\pi_{\mathcal{M},h}(a_h|s_h)}{\pi_{\mathrm{ref},h}(a_h|s_h)} + V_{\mathcal{M},h}(s_h), \quad \text{if } h \leq H - 1$$

$$u(s_H, a_H) = \eta \cdot \log \frac{\pi_{\mathcal{M},H}(a_H|s_H)}{\pi_{\mathrm{ref},H}(a_H|s_H)} + V_{\mathcal{M},H}(s_H). \tag{8}$$

Summing over $h \in [H]$, we have the following re-parameterization result:

$$u(s_H, a_H) = \eta \underbrace{\sum_{h=1}^{H} \log \frac{\pi_{\mathcal{M},h}(a_h|s_h)}{\pi_{\mathrm{ref},h}(a_h|s_h)}}_{\text{term } (A)} + \underbrace{V_{\mathcal{M},1}(s_1)}_{\text{term } (B)} + \underbrace{\sum_{h=1}^{H-1}\left[V_{\mathcal{M},h+1}(s_{h+1}) - \mathbb{E}_{o_h \sim \mathbb{P}_h(\cdot|s_h, a_h)} V_{\mathcal{M},h+1}(s_{h+1})\right]}_{\text{term } (C)}. \tag{9}$$

Here, term $(A)$ is similar to the single-turn case and term $(B)$ will be cancelled for the reward difference of two samples with the same prompt $s_1$. However, in practice, term $(C)$ is typically not feasible to directly compute as term $(C)$ is related to the randomness of the external environment.

For the focus of this work, i.e., the tool-integrated mathematical reasoning, luckily the code execution result is determined by the history (the codes written by the LLMs). This leads to term $(C) = 0$. Therefore, we can plug equation 9 into the maximum likelihood estimation of the utility function with a dataset $\mathcal{D}$ consisting of $(x, \tau^w, \tau^l)$, to get the following multi-turn DPO (M-DPO) loss:

$$\mathcal{L}_{\text{M-DPO}}(\theta) = -\sum_{(x, \tau^w, \tau^l) \in \mathcal{D}} \log \sigma\left(\eta \sum_{h=1}^{H}\left[\log \frac{\pi_{\theta,h}(a_h^w|s_h^w)}{\pi_{\mathrm{ref},h}(a_h^w|s_h^w)} - \log \frac{\pi_{\theta,h}(a_h^l|s_h^l)}{\pi_{\mathrm{ref},h}(a_h^l|s_h^l)}\right]\right). \tag{10}$$

Similarly, we can implement M-KTO under deterministic transition. We refer interested readers to Appendix A for the loss function details.

## 2.3 ONLINE ITERATIVE TRAINING

We now combine the planning algorithm M-DPO with the online iterative learning framework, as inspired by its great success in the single-turn case (Xiong et al., 2024; Guo et al., 2024b).

**Learning objective.** For a more comprehensive understanding of its statistical behavior, we will consider two different learning objectives. The first objective is a KL-regularized one:

$$\max_\pi \mathbb{E}_{x\sim d_0}\mathbb{E}_{a_h\sim\pi(\cdot|s_h),o_h\sim\mathbb{P}_h^*(\cdot|s_h,a_h)}\Big[u^*(x,y) - \eta\sum_{h=1}^H D_{\mathrm{KL}}\big(\pi(\cdot|s_h),\pi_0(\cdot|s_h)\big)\Big], \quad (11)$$

i.e., $\max_\pi J(\pi;\mathcal{M}^*,\pi_0)$ where $\mathcal{M}^* = (\mathcal{S},\mathcal{A},H,\mathbb{P}^*,d_0,u^*)$ is the groundtruth environment and $\pi_0$ is the initial SFT policy. This target is widely adopted in RLHF and requires us to search for the optimal policy only at a *fixed* KL ball centered at the SFT policy $\pi_0$. In contrast, the second one is the non-regularized target, i.e., directly optimizing the reward:

$$\max_\pi \mathbb{E}_{x\sim d_0}\mathbb{E}_{a_h\sim\pi(\cdot|s_h),o_h\sim\mathbb{P}_h^*(\cdot|s_h,a_h)}\big[u^*(x,y)\big]. \quad (12)$$

This target is the standard one in canonical RL studies (Sutton & Barto, 2018). One motivation for this target is that in the reasoning task, the reward function is more interpretable (e.g. final result checking) compared to the chat task.

**Algorithmic framework.** We present a general online iterative algorithmic framework in Algorithm 1. This framework is termed as *Online Iterative Multi-turn Gibbs Sampling from Human Feedback (M-GSHF)* because the optimal policy is a layer-wise Gibbs distribution that generalizes the result in Xiong et al. (2024). We now discuss some features of the framework as follows.

*Reference model choice for controlling regularization level.* We unify the two different learning targets in equation 11 and equation 12 by taking the reference model choice as a hyperparameter. First, if we fix the reference model as the initial policy, i.e., $\pi_{t,\mathrm{ref}} = \pi_0, \forall t \in [T]$, we always search the optimal policy within the KL ball centered at $\pi_0$, and thus optimize the KL-regularized target. In contrast, inspired by the mirror descent (Nemirovskij & Yudin, 1983), if we update the reference policy every iteration to be the policy learned in the last iteration, i.e., $\pi_{t,\mathrm{ref}} = \pi_{t-1}^1, \forall t \in [T]$, the cumulative update can make the model to move away from the original $\pi_0$ (while a constraint is made on the per-iteration update magnitude) and we thus optimize the non-regularized target in equation 12. See Figure 1 for an illustration.

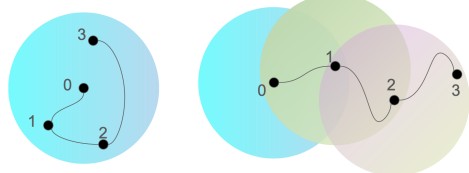

Figure 1: Illustration of the difference between the two learning objectives. Left: the KL-regularized target as we do not update the reference model. Right: the non-regularized target.

*Non-symmetric policy choice for exploration-exploitation trade-off.* We update our behavior policies in a non-symmetric way. The first agent aims to extract the historical information we have gathered so far and runs the M-DPO or M-DKO presented in Section 2.2. However, it is widely recognized in RL studies (Sutton & Barto, 2018; Auer et al., 2002) that simply exploiting the historical data via following the empirically best model is not sufficient to obtain a good final policy, while it is also required to explore the environment so that new information can be collected to facilitate subsequent learning, i.e., the exploration-exploitation tradeoff. Therefore, the second agent will strategically incorporate the uncertainty of the future relative to $\pi_t^1$ to choose $\pi_t^2$, which is referred to as the exploration policy.

A comprehensive theoretical analysis is derived for Algorithm 1, deferred to Appendix D due to space constraint, with a focus on the KL-regularized target. Here we highlight the following informal result (see Theorem 2 for the complete version), emphasizing the efficiency of Algorithm 1 guaranteed by a sublinear regret. The other target of optimizing the rewards has been theoretically studied in Wang et al. (2023b) while the techniques of analyzing mirror-descent-style algorithm have been developed in Cai et al. (2020).

**Theorem 1** (Informal). *Under the realizability assumption, with the KL-regularized target, the theoretical version of Algorithm 1 leads to a regret (defined in equation 15) that is sublinear in horizon $T$ for a broad class of reward and transition models.*

The main take-away message from the theorem is that if we choose suitable exploration policy, the online iterative learning is provably efficient. We also remark that without explicit mechanism to encourage exploration, the randomness of the LLM itself is not sufficient to learn the optimal policy (Zhang, 2022) if we do not make additional assumption.

Moving toward practical algorithm designs, the exploration is generally interpreted as increasing the diversity of the collected data by adopting inference-time methods with the base DPO policy $\pi_t^1$. For instance, one may tune the sampling temperature as in Llama project (Touvron et al., 2023) or use best-of-n sampling (Xu et al., 2023; Hoang Tran, 2024; Dong et al., 2024), where these methods outperform the vanilla on-policy sampling with considerable margin. In this work, we mainly enrich the generated data by various intermediate checkpoints, as done in the Claude project (Bai et al., 2022). We refer this approach as *mixture sampling*. It is also natural to adopt reward-guided Monte Carlo tree search (MCTS) (Xie et al., 2024b), which we leave for future work.

---

**Algorithm 1** Online Iterative M-GSHF

1: **Input:** KL coefficient $\eta > 0$, horizon $T > 0$, initial policy $\pi_0$, batch size $m > 0$.
2: Initialize $\mathcal{D} \leftarrow \emptyset$ and $\pi_1^1 = \pi_1^2 = \pi_{1,\text{ref}} \leftarrow \pi_0$.
3: **for** $t = 1, 2, \cdots, T$ **do**
4:      Sample $m$ pairs $(x, \tau^1, \tau^2, z)$ as $\mathcal{D}_t$ by $x \sim d_0, \tau^1 \sim \pi_t^1, \tau^2 \sim \pi_t^2$, receive the $m$ preference signals $z$ following the Bradley-Terry model from Definition 1 and update the preference dataset $\mathcal{D} \leftarrow \mathcal{D} \cup \mathcal{D}_t$.
5:      ▷ **Extract the empirically optimal policy from historical data**
6:      **Practical:** Perform the planning algorithms on $\mathcal{D}$ to get $\pi_t^1$ (e.g., using the M-DPO loss in equation 10 or the M-KTO loss in equation 13)
7:      **Theoretical:** Perform MLE on $\mathcal{D}$ to obtain model estimation $\hat{\mathcal{M}}_t = (\hat{u}_t, \hat{\mathbb{P}}_t)$ as in equation 16 and equation 17; call Oracle 2 with $\hat{\mathcal{M}}_t, \eta, \pi_{t,\text{ref}}$ to get $\pi_t^1$
8:      ▷ **Select the exploration policy to facilitate learning**
9:      **Practical:** Given $\pi_t^1$, select $\pi_t^2$ as an exploration policy using heuristic methods (such as mixture sampling, inference parameters tuning and west-of-n sampling.
10:      **Theoretical:** Given $\pi_t^1$, choose $\pi_t^2$ as an exploration policy following equation 18
11:      ▷ **Choose the reference model to control regularization level**
12:      Update $\pi_{t+1,\text{ref}} \leftarrow \pi_t^1$ when considering the non-regularized target; keep $\pi_{t+1,\text{ref}} \leftarrow \pi_0$ when considering the KL-regularized target
13: **end for**
14: **Output:** the best model in $\pi_{1:T}^1$ by a validation set.

---

## 3 EXPERIMENTS

### 3.1 EXPERIMENT SETUP

**Task, datasets, and models.** We use the test sets of MATH (Hendrycks et al., 2021) and GSM8K (Cobbe et al., 2021a) to measure the model's ability to solve the mathematical problems. To construct the training prompt set, we use the prompts from MetaMathQA (Yu et al., 2023) and MMIQC (Liu & Yao, 2024), which is an augmented prompt set from the 7.5K training problems of MATH and 7.47K training problems of GSM8K. We provide an example of the data sample in Figure 4. We train with a range of base models, including Gemma-1.1-it-7B (Team et al., 2024), CodeGemma-1.1-it-7B (Team, 2024), Mistral-7B-v0.3[2] (Jiang et al., 2023), and Gemma2-it-9B. We first fine-tune the model using a subset of the Open-MathInstruct dataset. The details of the SFT process are provided in Appendix B.

**Implementation of Iterative M-DPO and M-KTO.** We run the iterative training for 3 epochs in total. For each iteration, we have a prompt set of 20K questions and generate 20 responses per prompt with current DPO model and 10 responses per prompt with the model from last iteration.

---

[2]We use the pre-trained version because the chat template of its instruct model from huggingface is not consistent with their own codebase.

We check the final answer of these responses to determine their correctness. Then, for each prompt, we randomly sample two responses with correct and incorrect final answers and add them into the training samples. Then, we train the model on the collected samples using the M-DPO/M-KTO loss. We also include an ablation of reference model choice. To implement the M-DPO, we simply set the labels of all the user-turn tokens to be -100 and mask the log-probability in the subsequent loss computation. We train the model for 1 epoch at most and tune the learning rate in {2e-7, 4e-7, 7e-7, 1e-6} with the first iteration of iterative training. Eventually, the learning rate of 4e-7 is used for Gemma-1.1 models and 2e-7 is used for Gemma-2 model and Mistral model. The global batch size is 32 with a warm-up step of 40. We evaluate the model every 50 training steps by the split prompt set. The hyper-parameters are of M-KTO are mostly the same as the M-DPO. We also set the $\lambda_+ = \lambda_- = 1$ following the original KTO paper (Ethayarajh et al., 2024).

**Baselines.** The existing literature mainly focuses on the synthetic data generation and SFT to teach the models to use the external tool. We use the results from Toshniwal et al. (2024) as baselines because we use the same SFT dataset so the results are generally comparable. For the CoT baselines, we use the Wizardmath models from Luo et al. (2023). We also include the reward ranked fine-tuning (RAFT) as a baseline (Dong et al., 2023), which is also known as rejection sampling fine-tuning (Touvron et al., 2023). Another baseline is the single-turn online iterative DPO and KTO (Rafailov et al., 2023; Ethayarajh et al., 2024), which ignore the problem structure (i.e., the external messages) and treat the trajectory as a whole. In implementation, it means that we do not mask the tokens of external messages.

Table 1: Main results of different methods on the test sets of GSM8K and MATH. †: the model serves as the starting checkpoint of other methods. The results of the CoT methods are borrowed from the technical reports (Toshniwal et al., 2024; Gou et al., 2023b). For iterative M-DPO/M-KTO, we update the reference model by default if not specified. The gains relative to the SFT starting checkpoint are marked by ↑.

| Base Model | Method | with Tool | GSM8K | MATH | AVG |
|---|---|---|---|---|---|
| WizardMath-7B | SFT for CoT | ✗ | 54.9 | 10.7 | 32.8 |
| WizardMath-13B | SFT for CoT | ✗ | 63.9 | 14.0 | 39.0 |
| WizardMath-70B | SFT for CoT | ✗ | 81.6 | 22.7 | 52.2 |
| CodeLLaMA-2-7B | SFT | ✓ | 75.9 | 43.6 | 59.8 |
| CodeLLaMA-2-13B | SFT | ✓ | 78.8 | 45.5 | 62.2 |
| CodeLLaMA-2-34B | SFT | ✓ | 80.7 | 48.3 | 64.5 |
| CodeLLaMA-2-70B | SFT | ✓ | 84.6 | 50.7 | 67.7 |
| Gemma-1.1-it-7B | SFT† | ✓ | 77.5 | 46.1 | 61.8 |
| Gemma-1.1-it-7B | RAFT | ✓ | 79.2 | 47.3 | 63.3 |
| Gemma-1.1-it-7B | Iterative Single-turn DPO | ✓ | 81.7 | 48.9 | 65.3 |
| Gemma-1.1-it-7B | Iterative Single-turn KTO | ✓ | 80.6 | 49.0 | 64.8 |
| Gemma-1.1-it-7B | Iterative M-DPO + fixed reference | ✓ | 79.9 | 48.0 | 64.0 |
| Gemma-1.1-it-7B | M-DPO Iteration 1 | ✓ | 81.5 | 49.1 | 65.3 |
| Gemma-1.1-it-7B | M-DPO Iteration 2 | ✓ | 82.5 | 49.7 | 66.1 |
| Gemma-1.1-it-7B | M-DPO Iteration 3 | ✓ | **83.9** ↑6.4 | **51.2** ↑5.1 | **67.6** ↑5.8 |
| Gemma-1.1-it-7B | Iterative M-KTO | ✓ | 82.1 ↑4.6 | 49.5 ↑3.4 | 65.8 ↑4.0 |
| CodeGemma-1.1-it-7B | SFT† | ✓ | 77.3 | 46.4 | 61.9 |
| CodeGemma-1.1-it-7B | RAFT | ✓ | 78.8 | 48.4 | 63.6 |
| CodeGemma-1.1-it-7B | Iterative Single-turn DPO | ✓ | 79.1 | 48.9 | 64.0 |
| CodeGemma-1.1-it-7B | Iterative Single-turn KTO | ✓ | 80.2 | 48.6 | 64.4 |
| CodeGemma-1.1-it-7B | Iterative M-DPO | ✓ | 81.5 ↑4.2 | **50.1** ↑3.7 | **65.8** ↑4.0 |
| CodeGemma-1.1-it-7B | Iterative M-KTO | ✓ | **81.6** ↑4.3 | 49.6 ↑3.2 | 65.6 ↑3.8 |
| Mistral-7B-v0.3 | SFT† | ✓ | 77.8 | 42.7 | 60.3 |
| Mistral-7B-v0.3 | RAFT | ✓ | 79.8 | 43.7 | 61.8 |
| Mistral-7B-v0.3 | Iterative Single-turn DPO | ✓ | 79.8 | 45.1 | 62.5 |
| Mistral-7B-v0.3 | Iterative Single-turn KTO | ✓ | 81.3 | 46.3 | 63.8 |
| Mistral-7B-v0.3 | Iterative M-DPO | ✓ | **82.3** ↑4.5 | **47.5** ↑4.8 | **64.9** ↑4.7 |
| Mistral-7B-v0.3 | Iterative M-KTO | ✓ | 81.7 ↑3.9 | 46.7 ↑4.0 | 64.2 ↑4.0 |
| Gemma-2-it-9B | SFT† | ✓ | 84.1 | 51.0 | 67.6 |
| Gemma-2-it-9B | RAFT | ✓ | 84.2 | 52.6 | 68.4 |
| Gemma-2-it-9B | Iterative Single-turn DPO | ✓ | 85.2 | 53.1 | 69.2 |
| Gemma-2-it-9B | Iterative Single-turn KTO | ✓ | 85.4 | 52.9 | 69.2 |
| Gemma-2-it-9B | Iterative M-DPO | ✓ | **86.3** ↑2.2 | **54.5** ↑3.5 | **70.4** ↑2.9 |
| Gemma-2-it-9B | Iterative M-KTO | ✓ | 86.1 ↑2.0 | **54.5** ↑3.5 | 70.3 ↑2.8 |

## 3.2 Main Results

We evaluate the models in the zero-shot setting and report the main results in Table 1. From the first two sections in Table 1, we first observe that the tool-integrated LLMs significantly outperform their CoT counterparts with only SFT, demonstrating the benefits of leveraging external tools. In the subsequent discussions, we focus on the comparison within the scope of tool-integrated LLMs.

**Iterative M-DPO and M-KTO considerably improve the SFT models.**   Across all four base models, iterative training with M-DPO or M-KTO consistently leads to notable improvements over the initial SFT checkpoint on both GSM8K and MATH. In particular, with M-DPO, the aligned Gemma-1.1-it-7B model attains accuracies of 83.9% and 51.2% on GSM8K and MATH, respectively, and is comparable to the open-source Open-MathInstruct-finetuned CodeLLaMA-2-70B (slightly worse on GSM8K but also slightly better on MATH). Moreover, the aligned Gemma-2-it-9B model achieves accuracies of 86.3% and 54.5% on GSM8K and MATH, surpassing all of the open-source models trained with Open-MathInstruct in the 7B to 70B range. Overall, our framework can robustly further boost the tool-integrated models' ability after SFT.

**Iterative M-DPO and M-KTO surpass existing RLHF baselines.**   We also observe that the iterative M-DPO and M-KTO surpass other existing RLHF baselines. First, they consistently and significantly outperform RAFT across all four base models. This is because RAFT only imitates the correct trajectories, while the DPO-based and KTO-based algorithms further use the negative signal from incorrect trajectories. We note that the SFT stage in our pipeline can also be viewed as an application of RAFT. Consequently, our results should be interpreted to be that after the first stage of SFT, algorithms with negative signal are more sample efficient. Moreover, while the online iterative single-turn DPO (KTO) also gives a better performance, it is generally worse than the multi-turn version. This suggests that learning to predict the off-policy external messages returned by the code interpreter usually has a negative impact on the reasoning ability improvement. We also present a representative example we encounter in Figure 5, where LLMs generate poorly constructed code resulting in anomalous and lengthy external messages. Forcing LLMs to learn to predict these messages can significantly hurt the model's reasoning abilities.

**Iterative training and reference update lead to better performance.**   Using Gemma-1.1-it-7B with M-DPO as an example, we observe that online iterative training leads to better results. The GSM8K test accuracy increases from 77.5% (SFT) to 81.5% (iter 1) to 82.5% (iter2) to 83.9% (iter3), and the test accuracy of MATH improves from 46.1% (SFT) to 49.1% (iter 1) to 49.7% (iter2) to 51.2% (iter3). This aligns with our theoretical insight that iterative training helps models progressively explore and learn the optimal policy. Additionally, if the reference model remains fixed at the SFT policy, the final performance is notably worse compared to updating the reference model at each iteration. This likely occurs because the algorithm, in this case, optimizes the non-regularized reward, and the rewards in mathematical reasoning tasks are more accurate than in general chat tasks, leading to better in-domain performance. A detailed ablation study on the impact of KL regularization is deferred to the next section.

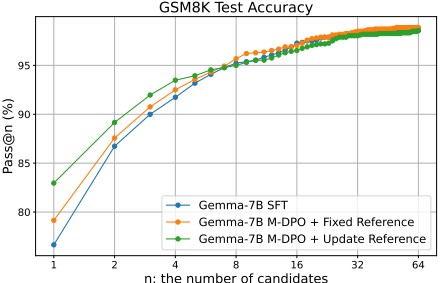
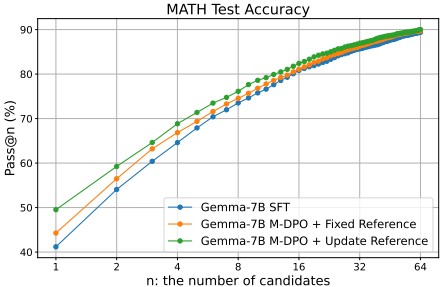

Figure 2: The pass@n rate with respect to the number of candidates n. We evaluate the models using temperature 0.7 following the previous works Shao et al. (2024); Toshniwal et al. (2024). We notice that preference learning only improves the metric pass@n when n is relatively small.

**Preference learning improves pass@n only when n is relatively small.** We plot the pass@n accuracy in terms of the number of candidate trajectories n in Figure 2. A question is solved if at least one of the n sampled trajectories is correct. We find that preference learning improves pass@n accuracy only when n is small. For $n > 16$, all models perform similarly on GSM8K and MATH, indicating that iterative M-DPO does not introduce new knowledge but instead enhances the quality of top-n responses. This observation also aligns with the result of CoT reasoning (Shao et al., 2024).

### 3.3 ABLATION STUDY AND DISCUSSION

**Moderate KL regularization balances per-iteration improvement and exploration.** The effectiveness of iterative DPO is highly dependent on the reference model and KL coefficient. In our ablation study, we first consider two different choices of the reference model: (1) using the fixed reference model $\pi_0$; (2) updating the reference model to the last iteration's model at each round, which can be viewed as a trade-off between the generation diversity and reward optimization. As shown in Table 3.3, models with an updated reference

| Method | GSM8K | MATH |
|---|---|---|
| SFT | 77.5 | 46.1 |
| update reference + $\eta = 0.01$ | 81.7 | 50.1 |
| update reference + $\eta = 0.1$ | 83.9 | 51.2 |
| update reference + $\eta = 0.5$ | 82.8 | 49.7 |
| fixed reference + $\eta = 0.1$ | 79.9 | 48.0 |

Table 2: Ablation study of the impact of KL regularization on iterative M-DPO.

model outperform those with a fixed reference model. We hypothesize that in reasoning tasks, the correct reasoning paths are highly concentrated, making diversity less crucial so optimizing the non-regularized reward gives superior model performance.

Previous work (Tunstall et al., 2023) on offline DPO suggests that a lower KL coefficient (0.01) improves performance by allowing the model to deviate more from the SFT model $\pi_0$. In our ablation study, we search the KL coefficient $\eta \in \{0.01, 0.1, 0.5\}$. According to Table 3.3, we find that the strongest model is obtained by a moderate KL coefficient of $0.1$, outperforming both 0.01 and 0.5. To explain this, we plot the GSM8K test accuracy (Figure 3) during iterative training. In the first iteration, lower KL values show larger improvements, consistent with Tunstall et al. (2023)'s results. However, models trained with very low KL coefficients lose diversity quickly, reducing their ability to generate diverse trajectories for later training, leading to diminishing returns in subsequent iterations. Conversely, a higher KL coefficient of 0.5 imposes too much regularization, limiting improvement per iteration. In summary, for online iterative training, a balance between per-iteration improvement and exploration efficiency is key to optimizing overall performance, an intuition that also applies to sampling strategies and other experimental techniques.

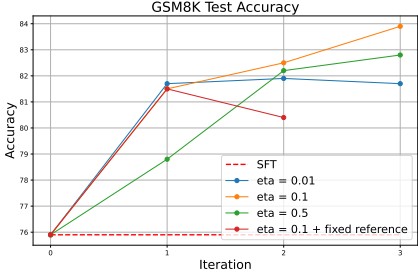 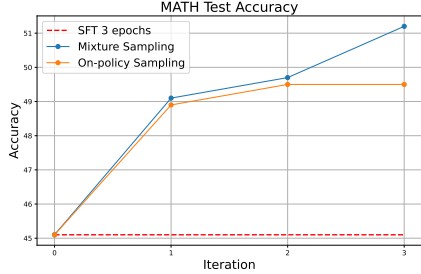

Figure 3: Left: the test accuracy on GSM8K dataset with different levels of KL regularization. Right: the test accuracy on MATH dataset with different sampling strategies.

**The impact of sampling strategy: data diversity and coverage are crucial.** During iterative training of Gemma-1.1-it-7B, we see an increase in correct trajectories from 47% in the first iteration to 76% in last iteration. Moreover, as the reference model updates at each step, trajectory diversity declines, which is critical for DPO/KTO training due to its contrastive nature. We follow Bai et al. (2022); Dong et al. (2024) to explore two data collection strategies: (1) on-policy sampling (trajectories sampled from the current model) and (2) mixture sampling (20 trajectories from the current model and 10 from model of previous iteration). As shown in Table 6, mixture sampling significantly outperforms on-policy sampling, particularly in the third iteration where on-policy sam-

pling fails to improve MATH test accuracy. This highlights the importance of diversity in iterative training and aligns with previous findings that advanced exploration strategies help prevent diversity collapse and improve preference learning (Bai et al., 2022; Touvron et al., 2023; Xiong et al., 2024; Pace et al., 2024; Dong et al., 2024). It would also be interested to explore more advanced exploration strategy like MCTS in the future study.

To ensure both correct and incorrect reasoning paths exist, we collected N trajectories per prompt. A larger N generally improves prompt coverage, as more samples are needed for difficult problems. For example, in iteration 1, with N=30, 92.5% of the prompts are covered, compared to 83.0% for N=12 and 60% for N=6. See Figure 2 for an illustration of the relationship between pass@1 and N. However, increasing N also increases computational costs. In our ablation study (Table 3.3), we find that increasing N

| Method | GSM8K | MATH |
|---|---|---|
| SFT | 77.5 | 46.1 |
| N=30 + Mixture | 83.9 | 51.2 |
| N=12 + Mixture | 83.5 | 51.2 |
| N=6 + Mixture | 82.0 | 49.2 |
| N=30 + On-policy | 83.1 | 49.5 |

Table 3: Ablation study of the sampling strategy with iterative M-DPO and Gemma-1.1-it-7B.

from 6 to 12 leads to a significant performance boost, reflecting better coverage for complex problems. However, increasing N from 12 to 30 yields only minor improvements, suggesting that the benefits of larger N diminish quickly in vanilla rejection sampling. We expect that difficulty-aware sampling can lead to a better performance, while maintaining a moderate inference cost.

## 4 CONCLUSION, LIMITATION, AND FUTURE RESEARCH DIRECTION

In this paper, we demonstrate that preference learning, as an alternative to supervised fine-tuning, further enhances the performance of tool-integrated reasoning LLMs after SFT. We introduce an online iterative multi-turn direct preference optimization algorithm, validated through extensive experiments across multiple base models. Results show significant improvements in pass@1 over the SFT policy, particularly on benchmarks like GSM8K and MATH. Ablation studies highlight the importance of balancing per-iteration improvement with exploration, which is achieved by moderate levels of KL regularization and strategic exploration choices.

Several avenues for improvement remain unexplored. Our current approach only uses final result checks as preference signals, limiting the comparison between trajectories with correct or incorrect answers. One may use step-wise reward signal (Lightman et al., 2023) in the data ranking stage. Meanwhile, the fine-grained reward signals could enable the use of advanced exploration strategies like west-of-n sampling (Pace et al., 2024), or MCTS (Xie et al., 2024b) in our heuristic exploration implementation. Finally, while the direct preference learning algorithms show promising gains for the mathematical reasoning tasks with code interpreter, it is not directly applicable to the general agent learning with more complex and stochastic external environments or against dynamic opponents. In particular, it requires to construct a value network for involving an adaptive margin in the optimization target and take the randomness of the external environment into consideration. We leave the study of this more involved algorithm to the future work.

### ACKNOWLEDGEMENTS

Wei Xiong and Tong Zhang are partially supported by an NSF IIS grant No. 2416897

### REPRODUCIBILITY STATEMENT

We believe that making the result reproducible is important. Following the author guidance of ICLR, we present a reproducibility statement here to help the interested readers to reproduce our result. Most implementation details, including hyperparameters, are provided in Section 3.1 and Appendix B. Additionally, we have open-sourced our training code along with a step-by-step guide, using Gemma-1.1-it-7B as an example. We have also made the processed SFT dataset, prompt set, and the training data for the first iteration of M-DPO/M-KTO available for easy download (see supplemental materials for details). The RLHF experiments of this paper are run with 8xA100 80G GPUs, where an additional machine with 8xA100 40G GPUs is also used to accelerate data

collection and model evaluation. The main experiment of this paper can be reproduced within 24 - 48 hours with this setup. To improve the readability of this paper, we provide a notation table in Appendix A. The informal version of our main theoretical result is summarized in Theorem 1 is re-stated in Theorem 2 and its proof is provided in Appendix D.

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

# A  NOTATION TABLE, RELATED WORK, AND MISSING DETAILS

## A.1  NOTATION TABLE

| Notation | Description |
|---|---|
| $x, \mathcal{X}$ | The prompt and the prompt space. |
| $d_0$ | The distribution of initial state (prompt). |
| $s_h \in \mathcal{S}, a_h \in \mathcal{A}, o_h$ | The state, action, and observation. |
| $H$ | Episode length, e.g., the maximal number of tool calls. |
| $\mathbb{P}^* = [\mathbb{P}_h^*]_{h=1}^H$ | The true observation kernel. |
| $\tau = (x, y)$ | $\tau$ is a trajectory and $y$ is the completion part, i.e., we exclude $x$ from $\tau$. |
| $u^*$ | The true utility function associated with the BT model defined in Definition 1. |
| $\mathcal{M}^* = (\mathcal{S}, \mathcal{A}, H, \mathbb{P}^*, d_0, u^*)$ | The true model with observation kernel $\mathbb{P}^*$ and utility function $u^*$ |
| $\sigma(\cdot)$ | $\sigma(z) = 1/(1 + \exp(-z))$ is the sigmoid function. |
| $z \in \{0, 1\}$ | Preference signal. |
| $\pi = [\pi_h]_{h=1}^H$ | The policy, which is parameterized by the LLM. |
| $\mathcal{M} = (\mathcal{S}, \mathcal{A}, H, \mathbb{P}, d_0, u)$ | One arbitrary environment with observation kernel $\mathbb{P}$ and utility function $u$. |
| $\pi_{\mathrm{ref}} = [\pi_{\mathrm{ref},h}]_{h=1}^H$ | One arbitrary reference policy. |
| $J(\pi; \mathcal{M}, \pi_{\mathrm{ref}})$ | The KL-regularized target (equation 2) with environment $\mathcal{M}$ and reference $\pi_{\mathrm{ref}}$. |
| $\eta$ | The coefficient of KL penalty, defined in equation 2. |
| $Q_{\mathcal{M}} = [Q_{\mathcal{M},h}]_{h=1}^H$ | The optimal $Q$-values associated with $J(\pi; \mathcal{M}, \pi_{\mathrm{ref}})$, defined in equation 3. |
| $V_{\mathcal{M}} = [V_{\mathcal{M},h}]_{h=1}^H$ | The optimal $V$-values associated with $J(\pi; \mathcal{M}, \pi_{\mathrm{ref}})$, defined in equation 4. |
| $\pi_{\mathcal{M}} = [\pi_{\mathcal{M},h}]_{h=1}^H$ | The optimal policy associated with $J(\pi; \mathcal{M}, \pi_{\mathrm{ref}})$, defined in equation 4. |
| $\mathcal{L}_{\text{M-DPO}}(\cdot)$ | M-DPO loss, defined in equation 10. |
| $\mathcal{L}_{\text{M-KTO}}(\cdot)$ | M-KTO loss, defined in equation 13. |
| $J(\pi)$ | The abbreviation of $J(\pi; \mathcal{M}^*, \pi_0)$, defined in equation 14. |
| $\pi^* = [\pi_h^*]_{h=1}^H$ | The optimal policy associated with $J(\pi)$. |
| $\pi_t^1, \pi_t^2$ | The main and exploration policy at round $t$ |
| $\mathrm{Reg}(T)$ | Regret over horizon $T$, defined in equation 15. |
| $\mathcal{U}, \mathcal{P}$ | Known sets such that $u^* \in \mathcal{U}$ and $\mathbb{P}^* \in \mathcal{P}$ |
| $B$ | Assuming $u^*(x, y) \in [0, B], \forall(x, y)$. |
| $\hat{u}_t, \hat{\mathbb{P}}_t$ | MLE of $u^*$ and $\mathbb{P}^*$ at round $t$, defined in equation 16 and equation 17. |
| $\widetilde{\mathcal{U}}_t, \widetilde{\mathcal{P}}_t$ | Confidences sets of $u^*$ and $\mathbb{P}^*$ at round $t$, defined in equation 19. |
| $c_1, c_2, c$ | Absolute constants. |
| $\kappa$ | $1/(2 + \exp(-B) + \exp(B))$. |
| $d_{\mathcal{U}}$ | Eluder coefficient from Definition 3. |
| $d_{\mathcal{P}}, \xi(\cdot)$ | Generalized Eluder-type condition from Definition 4. |
| $\mathrm{TV}(\cdot, \cdot)$ | Total variation distance between two distributions. |

Table 4: The table of notations used in this paper.

## A.2  RELATED WORK

**LLMs for Mathematical Problem Solving.**  A line of works proposes to prompt LLMs to solve the complex reasoning task in a step-by-step manner, known as the Chain-of-Thought (CoT) prompting (Wei et al., 2022; Zhou et al., 2022; Zhu et al., 2022; Tong et al., 2024), which has been a standard practice in reasoning task. However, LLMs often struggle with basic arithmetic and symbolic manipulations when relying solely on internal knowledge and natural language reasoning, as measured by standard benchmarks (Cobbe et al., 2021a; Hendrycks et al., 2021). To overcome these limitations, several studies have explored the use of external tools to enhance the LLMs' problem-solving abilities. This includes calculators (Cobbe et al., 2021b; Shao et al., 2022), symbolic solvers (Zhang, 2023), and code interpreters (Mishra et al., 2022; OpenAI, 2023). A particularly effective approach is the Program-based method (PoT), which performs CoT reasoning by writing code and using the output of the written code as the final answer (Gao et al., 2023; Chen et al., 2022). This method significantly outperforms traditional CoT-based techniques in mathematical problem solving. However, PoT also faces challenges in planning and error handling, where natural language reasoning is more suitable (Gou et al., 2023a). In view of this, tool-integrated reasoning is proposed to combine the natural-language-based intrinsic reasoning with the external tools (Gou et al., 2023b) and has achieved great progresses in recent studies (Gou et al., 2023b; Yue et al., 2023; Yu et al., 2023;

Shao et al., 2024; Toshniwal et al., 2024). While these efforts have primarily focused on synthetic data generation for tool-integrated reasoning, our work aims to further boost the performance of tool-integrated LLMs by RLHF.

**RLHF and RLHF Algorithms.** The predominant approach in RLHF is the deep RL method, Proximal Policy Optimization Algorithms (PPO) (Schulman et al., 2017), which leads to the great successes in Chat-GPT (OpenAI, 2023), Gemini (Gemini et al., 2023), and Claude (Anthropic, 2023). However, applying PPO requires extensive efforts and resources (Choshen et al., 2019; Engstrom et al., 2020), often beyond the scope of open-source capabilities. In view of this, alternative approaches have been developed. The rejection sampling fine-tuning was first proposed with the name RAFT (reward ranked fine-tuning) in RLHF (Dong et al., 2023) and was later extended to machine translation (Gulcehre et al., 2023) and mathematical problem solving (Yuan et al., 2023a). Its theoretical advantage was explored in Gui et al. (2024). Subsequently, another long line of works proposes direct preference learning algorithms, including SLiC (Zhao et al., 2023), DPO (Rafailov et al., 2023), IPO (Azar et al., 2023), KTO (Ethayarajh et al., 2024), and GPO (Tang et al., 2024). These algorithms bypass the reward modeling step and optimize carefully designed loss objectives directly on the preference dataset, hence the name direct preference learning. There are also some works focusing on more general preference structure Munos et al. (2023); Swamy et al. (2024); Ye et al. (2024); Rosset et al. (2024) beyond the reward-based framework or post-processing of the model (Lin et al., 2023; Zheng et al., 2024).

The newly proposed direct preference learning algorithms have largely advanced the RLHF area, particularly the post-training of open-source models, with the Zephyr project as a notable example (Tunstall et al., 2023). After this, a long line of work (e.g., Liu et al., 2023b; Xiong et al., 2024; Guo et al., 2024b; Xu et al., 2023; Tajwar et al., 2024; Xie et al., 2024a; Zhang et al., 2024b; Liu et al., 2024a;b; Meng et al., 2024) demonstrates the effectiveness of on-policy sampling (the samples are generated by the policy to be trained) and online exploration in enhancing direct preference learning. In particular, the online iterative DPO (Xiong et al., 2024; Xu et al., 2023; Hoang Tran, 2024) and its variants (e.g., Chen et al., 2024b; Rosset et al., 2024; Cen et al., 2024; Zhang et al., 2024c) have made state-of-the-art open-source models (Dong et al., 2024), or even the industry models (qwe, 2024; Meta, 2024). Despite these advancements, most algorithms are proposed and designed for single-turn interactions and chat. The scenarios beyond single-turn chat remain largely unexplored in the existing literature. One exception is the very recent work by Shani et al. (2024), which studies multi-turn chat task under general preferences. In contrast, in this paper, we aim to explore the use of RLHF in multi-turn tasks that incorporate interactions with external tools. Meanwhile, they derive a mirror-descent-based policy optimization algorithm, which is also different from ours.

**RLHF for Math Problem Solving.** Algorithms traditionally used in general chatbot applications have been adapted to enhance the reasoning capabilities of LLMs in mathematical contexts. For instance, RAFT (Reward-rAnked Fine-Tuning) (Dong et al., 2023; Yuan et al., 2023b; Touvron et al., 2023) is extensively employed for synthetic data generation, whether through on-policy (self-improving) (Yuan et al., 2023a) or off-policy (knowledge distillation) methods (Gou et al., 2023b; Yu et al., 2023; Toshniwal et al., 2024; Singh et al., 2023; Tong et al., 2024). The reward signal in these scenarios is typically derived from either final result checking or Outcome-supervised Reward Models (ORMs) (Uesato et al., 2022; Zelikman et al., 2022). A novel approach by Lightman et al. (2023) introduces Process-supervised Reward Models (PRMs), which provide feedback at each step of the Chain-of-Thought, demonstrating significant improvements over ORMs when combined with rejection sampling (Lightman et al., 2023; Wang et al., 2023a).

In addition to the RAFT, the GRPO algorithm proposed in Shao et al. (2024) studies multi-turn math problem solving but focuses on the CoT format without external inputs and the resulting model achieves the state-of-the-art performance in its class. The GRPO is a variant of Reinforce (Williams, 1992) thus falling into the scope of deep RL methods.

Further advancements include adapting direct preference learning algorithms to mathematical problem solving. For instance, Jiao et al. (2024); Yuan et al. (2024) have applied the original DPO or KTO by taking the trajectory completion as a "meta" action. Xie et al. (2024b); Pang et al. (2024) further adapt the online iterative DPO originally designed for chat (Xiong et al., 2024; Xu et al., 2023; Hoang Tran, 2024) and achieve better performance for CoT reasoning. Inspired by the success of PRMs, recent studies have explored generating proxy step-wise labels for the intermediate

steps of the reasoning trajectories. For instance, Xie et al. (2024b); Chen et al. (2024a); Lai et al. (2024) leverage Monte Carlo Tree Search (MCTS) and use the estimated Q value to generate the proxy labels for the intermediate steps. Lai et al. (2024) proposes to use AI feedback like GPT-4 (Lai et al., 2024) to find the first error step in the trajectory. Meanwhile, Lu et al. (2024) identifies a trajectory with the correct final answer and no errors as preferable, and prompts the SFT model with a high temperature, starting from some intermediate step to collect a rejected trajectory with errors (Pi et al., 2024). Finally, a very recent study by Chen et al. (2024a) proposes to use MCTS with a backward iteration from the final leaf node to compute the proxy unregularized value of each node. Preference pairs are then extracted from the tree by fixing the prefix and comparing *the next single reasoning step*. Then, they run the original DPO on these intermediate actions with the proxy labels from MCTS. To summarize, these works present different ways of preference data collection and apply the original DPO algorithm (with some additional marginal loss and regularization adapted from the literature), thereby differing from our work in both algorithmic concepts and application scope. In contrast, we study preference learning in the context of trajectory-level comparison, where we derive the optimality condition and introduce a multi-turn DPO within an online iterative framework, specifically for tool-integrated mathematical problem solving. However, we remark that while we focus on the trajectory-level comparison, the preference signal itself can be generated in a step-by-step supervision (see Section 2.1 for the detailed examples). When preference signals for partial trajectories with shared prefixes are available, our method can also adapt to learn these step-level signals (see the optimality condition in equation 9). In particular, the algorithmic design presented in this paper can be readily combined with the MCTS-based data collection strategy outlined in recent literature, which we leave for future work.

## A.3 MISSING DETAILS

**Multi-turn KTO.** With equation 9 implying that with term $(C) = 0$, the implicit reward is given by $A = \eta \sum_{h=1}^{H} \log \frac{\pi_h^*(a_h|s_h)}{\pi_{\text{ref},h}(a_h|s_h)}$, a multi-turn version of KTO (Ethayarajh et al., 2024), denoted as M-KTO, can also be naturally derived:

$$\mathcal{L}_{\text{M-KTO}}(\theta) = \mathbb{E}_{x,y \sim \mathcal{D}}\big[\lambda_y - v(x,y)\big], \tag{13}$$

where

$$u_\theta(x,y) = \eta \sum_{h=1}^{H} \log \frac{\pi_{u,h}(a_h|s_h)}{\pi_{\text{ref},h}(a_h|s_h)},$$

$$z_0 = \mathbb{E}_{x' \sim \mathcal{D}, \tau' \sim \pi_\theta(\cdot|x')} \sum_{h=1}^{H} D_{\text{KL}}\big(\pi_\theta(\cdot|s_h), \pi_{\text{ref}}(\cdot|s_h)\big),$$

and

$$v(x,y) = \begin{cases} \lambda_+ \sigma\big(\eta(u_\theta(x,y) - z_0)\big) & \text{if } y \sim y_{desirable}|x \\ \lambda_- \sigma\big(\eta(z_0 - u_\theta(x,y))\big) & \text{if } y \sim y_{undesirable}|x \end{cases}.$$

Here $\lambda_+$ and $\lambda_-$ are two hyper-parameters. We notice that Mitra et al. (2024) developed an online iterative version of KTO for the CoT format reasoning task. Here we extend it to build the tool-integrated reasoning agent.

## B IMPLEMENTATION DETAILS

**Supervised fine-tuning (SFT).** We first fine-tune the model for the tool-integrated reasoning task (Gou et al., 2023b), using a subset of the Open-MathInstruct dataset, which was generated by the permissively licensed `Mixtral-8x7B` model through in-context learning. The problems are from the training sets of MATH and GSM8K datasets. We restrict the number of samples for each question to be 50 and remove the nearly duplicate responses. Eventually we get 510K samples in the SFT dataset. We train the models for 4 epochs at most with a learning rate of 5e-6 for Gemma instruct models (Team et al., 2024) and a learning rate of 1e-5 for Mistral-v0.3 model (Jiang et al., 2023). The learning rates are determined by searching {2e-6, 5e-6, 1e-5}. We use a cosine learning rate scheduler and set the warm-up steps as 100. The samples are packed into blocks with length 4096 to accelerate training and a global batch size of 64 is used. We also mask all the user messages (i.e., the

---

**Multi-turn Reasoning with Python Interpreter for Mathematical Problems**

*Prompt*:
**User:** Let

$$f(x) = \begin{cases} ax + 3, & \text{if } x > 2, \\ x - 5 & \text{if } -2 \leq x \leq 2, \\ 2x - b & \text{if } x < -2. \end{cases}$$

Find $a + b$ if the piecewise function is continuous (which means that its graph can be drawn without lifting your pencil from the paper) $(s_1 = x)$
**Assistant:** The function must be continuous at 2 and $-2$. That means $x - 5 = ax + 3$ at $x = 2$ and $x - 5 = 2x - b$ at $x = -2$. Let's now solve for $x$ in each case. $(f_1)$

```python
from sympy import symbols, Eq, solve
## define the variables
x, a, b = symbols('x a b')

## the function must be continuous at 2
eq1 = Eq(x-5, a*x + 3)
eq1_sol = solve(eq1, x)

## the function must be continuous at -2
eq2 = Eq(x-5, 2*x - b)
eq2_sol = solve(eq2, x)

## print the solutions
print(eq1_sol)
print(eq2_sol)
```

$(e_1)$

**User:** "output [-8/(a - 1)] [b - 5]" $(o_1)$
**Assistant:** Thus, $x = -8/(a - 1)$ when $x = 2$ and $x = b - 5$ when $x = -2$. We can now solve for $a$ and $b$. $(f_2)$

```python
## x = -8/(a-1) when x = 2
eq3 = Eq(-8/(a-1), 2)
a = solve(eq3, a)
print(a)

## x = b-5 when x = -2
eq4 = Eq(b-5, -2)
b = solve(eq4, b)
print(b)
```

$(e_2)$
**User:** "output [-3]
[3]" $(o_2)$
**Assistant:** Thus, $a = -3$ and $b = 3$. Hence, $a + b = \boxed{0}$. $(f_3)$

---

Figure 4: An example of multi-turn mathematical reasoning with Python interpreter. The action is in a ReAct style (Yao et al., 2022) where it consists of a reasoning step $f_h$ and an execution step $e_h$.

prompt and the messages returned by the Python interpreter) in the training. The checkpoint at the end of the third epoch is used for Gemma and the checkpoint of the end of the second epoch is used for Mistral as the starting point for RLHF. This is because these models outperform the last-iteration one with considerable margin and is very close to the next one. An ablation study on the SFT epochs is also included.

**Data format and generation.** We format the data into a multi-turn chat where the user asks the LLMs a question, and provide the messages returned by the Python interpreter in the subsequent user rounds of chat. For all the data generation process, we adopt the following constraints: (1) for each turn, the model can generate up to 512 tokens; (2) the maximal number of steps is H=6; (3) the maximal number of token for each trajectory is 2048. Following Gou et al. (2023b); Toshniwal et al. (2024), the LLM agent is allowed to call the python interpreter when it decodes a python code

starting with ```python and ending with ```. For each step $h$, to generate the observation $o_h$, we leverage the python package `IPython`, and run all the codes in the history one by one and treat each code snippet as a Jupyter cell. We only return the standard output or the error message from the last snippet. When there exists some bug in the code, we only return the error message which is typically less than 20 tokens as in Toshniwal et al. (2024). We notice that some works (e.g. Shao et al. (2024)) also returns the first and the last 50 tokens of the trackback information.

**Evaluation Configuration.** All the models are evaluated in the zero-shot setting. For all the data generation process, we adopt the following constraints: (1) for each turn, the model can generate up to 512 tokens; (2) the maximal number of steps is H=6; (3) the maximal number of generated token for each trajectory is 2048. When collecting new data for online iterative M-DPO, we set temperature to be 1.0 and decode without top-K or top-p sampling. For evaluation, greedy decoding is employed so that the results are generally comparable with previous works Gou et al. (2023b); Toshniwal et al. (2024). For evaluating the models with pass@n rate, we follow Toshniwal et al. (2024) to adopt a temperature of 0.7.

**Python Experiment Environment.** We find that the evaluation can be influenced by the python environment, the precision (especially for the Gemma-1.1 models), and even the virtual machine we use. This does not affect the overall trend and conclusion because the magnitude of oscillation is relatively small compared to the overall improvement. For completeness, however, we specify some of the key package versions here. We use transformers 4.42.4, torch 2.3.0, sympy 1.2, antlr4-python3-runtime 4.11.0, IPython 8.26.0 for all models. We evaluate the models using torch.float and use vllm 0.5.0.post1 for most the experiments except for Gemma-2 where vllm 0.5.1 is required. The inconsistency of vllm version is because Gemma-2 model was not released when we performed the main experiments of this project. We fix the python environment and machine for our evaluation throughout the experiment. For SFT, we use the open-source axolotl project with version 0.4.1 and for online iterative preference learning and RAFT, we use the code base from RLHF Workflow (Dong et al., 2024).

**RAFT implementation.** RAFT first collects N trajectories per prompt, filters the low-quality data (by reward function), and fine-tune on the selected trajectories. The data generation step is similar to the online iterative M-DPO training, except that we only keep the trajectories with correct final answer. For each prompt, we sample at most $k$ trajectories where we search $k \in \{1, 3, 8\}$ and use $k = 1$ eventually because we do not see improvement by leveraging more data. We run the algorithm for three iterations in total. The training parameters are similar to the SFT stage, but we use a smaller batch size of 32 so that there are enough optimization steps. For Gemma models, we use a learning rate of 5e-6. For each training stage, we train the models for two epochs in total according to our parameter search. For Mistral model, we find that a smaller learning rate of 1e-6 and training for 1 epoch give us much better performance.

**Prompt template.** We do not tune the prompt though we do observe that the prompt engineering can further improve the performance. For all the experiments, we simply adopt the chat template of the models to form a multi-turn chat as in Figure 4.

## C   ADDITIONAL EXPERIMENTAL RESULTS

We include additional ablation studies in this section for a more comprehensive understanding of the proposed algorithm.

**The best model is obtained with starting checkpoint fine-tuned with more than 1 epochs.** Tunstall et al. (2023) finds that if the SFT model is trained for more than one epoch, the subsequent DPO training will lead to performance regression with longer training in terms of instruction-following ability and benchmark for a general chatbot. In other words, there exists a trade-off between the SFT training epochs and the DPO training steps. Moreover, the best model is obtained by SFT for one epoch in their practice. We also conduct an ablation study on the impact of the SFT epoch and summarize the results in Table 5. Consistently across all tested scenarios, the subsequent iterative M-DPO training leads to considerable model improvement compared to the SFT model. Meanwhile,

---

**An example of external messages returned by the Python interpreter**

*Prompt*:
**User:** Convert the periodic decimal notation 0.42 followed by an infinite sequence of 42s into a reduced fraction $\frac{p}{q}$, such that $p$ and $q$ have no common factors other than 1. Find the sum of $p$ and $q$.

```python
1  import sympy as sp
2
3  # Define the number as a sympy number with infinite repeating digits
4  digits = '42'
5  num = sp.N(0.4242424242, 1000000)  # 1000000 digits are accurate
6  num
```

**User:** ""output 0.42424242420000000234043000091332942247390747070312500000000000000000000 (omit other 320 '0')
File "/site-packages/timeout_decorator/timeout_decorator.py", line 69, in handler_raise_exception(timeout_exception, exception_message)
File "../anaconda3/envs/inference/lib/python3.10/site-packages/timeout_decorator/timeout_decorator.py", line 45, in _raise_exception
    raise exception()
timeout_decorator.timeout_decorator.TimeoutError: 'Timed Out' ""

Figure 5: An example of external messages returned by the Python interpreter. The model writes down a bad python code leading to an anomalous and lengthy error message.

we also observe a similar trade-off between SFT and RLHF training because with more SFT epochs, the gains from the RLHF stage decrease. However, in our case, the strongest model is obtained with three epochs of SFT, followed by fine-tuning through iterative M-DPO, which is different from the offline DPO training (Tunstall et al., 2023) or the iterative DPO for general chatbot (Dong et al., 2024) with only one epoch of SFT.

Table 5: Ablation study of the impact of SFT epoch. Mixture sampling is adopted for the iterative M-DPO training and we run for three iterations in total. The gains relative to their starting SFT checkpoints are marked by ↑.

| Model | Method | GSM8K | MATH |
|-------|--------|-------|------|
| Gemma-1.1-it-7B | SFT 1 epoch | 75.1 | 41.1 |
| Gemma-1.1-it-7B | SFT 1 epoch + Iterative M-DPO | 80.6 ↑5.5 | 46.7 ↑5.6 |
| Gemma-1.1-it-7B | SFT 2 epoch | 75.3 | 44.0 |
| Gemma-1.1-it-7B | SFT 2 epoch + Iterative M-DPO | 82.4 ↑7.1 | 49.8 ↑5.8 |
| Gemma-1.1-it-7B | SFT 3 epoch | 77.5 | 46.1 |
| Gemma-1.1-it-7B | SFT 3 epoch + Iterative M-DPO | 83.9 ↑6.4 | 51.2 ↑5.1 |

**NLL loss helps when the SFT model is substantially underfitting.** The recent work Pang et al. (2024) has introduced iterative RPO, specifically aimed at enhancing Chain of Thought (CoT) capabilities for solving mathematical problems. A key feature of this approach is the inclusion of an additional negative log-likelihood (NLL) loss for the preferred response. The main intuition for adding the NLL loss is that the original DPO algorithm (Rafailov et al., 2023) tends to reduce the likelihood of the preferred responses, and this is believed to hurt the reasoning ability (Wang et al., 2024). Motivated by their results, we explored the applicability of this idea to our setup. We conduct an ablation study by adding the NLL loss into the iterative M-DPO training and observe performance regression as reported in Table 6. We observe that the best model is obtained in the second iteration if we add the additional NLL loss even though we use the mixture sampling to increase the diversity of the collected data. With time-weighted exponential moving average for smoothing training record, we observe that the log probability of the chosen responses and rejected responses are (-126, -222) at the 200th step of the third iteration training when we add the NLL loss, as compared to (-166, -350) in the case without the NLL loss. This is consistent with the result of Pang et al. (2024) where with the additional NLL loss, both the log probability of chosen responses and that of rejected responses

increase. These evidences indicate that the NLL loss further contributes to the model distribution collapse and eventually hurt the overall performance of online iterative learning. Finally, we notice that the additional NLL loss can be viewed as an implementation of the pessimistic principle (Liu et al., 2024b). This also explains its inferior in-domain performance though it may be helpful to stable the training, which requires more in-depth studies.

However, one distinct feature between our setup and Pang et al. (2024) is whether we first fine-tune the initialized SFT model with in-domain data. To further understand the phenomena, we fine-tune the Gemma-1.1-it-7B with only 100 steps (so that the model knows to leverage Python code to solve the problem) as the starting checkpoint of preference learning and conduct an ablation study with the NLL loss using this model. We observe when the SFT model is substantially underfitting, the addition of NLL loss actually enhances performance. This scenario mirrors the findings of Pang et al. (2024), who utilized a general LLaMA2-70B-chat model (Touvron et al., 2023) without firstly fine-tuning on the in-domain data. Our observations align with prior research in the context of developing general chatbots (Lin et al., 2023), which suggests that RLHF is less effective without preliminary SFT.

Table 6: Other ablation studies. Mixture sampling is adopted for the iterative M-DPO training and we run for three iterations in total. The gains relative to the iterative M-DPO are marked by ↑.

| Model | Method | GSM8K | MATH |
|---|---|---|---|
| Gemma-1.1-it-7B | SFT 3 epoch | 77.5 | 46.1 |
| Gemma-1.1-it-7B | SFT 3 epoch + Iterative M-DPO | 83.9 | 51.2 |
| Gemma-1.1-it-7B | Iterative M-DPO with NLL loss | 81.7 ↓2.2 | 49.5 ↓1.7 |
| Gemma-1.1-it-7B | SFT 100 steps | 50.8 | 23.7 |
| Gemma-1.1-it-7B | + M-DPO Iteration 1 | 57.8 | 27.9 |
| Gemma-1.1-it-7B | + M-DPO and NLL loss Iteration 1 | 61.0 ↑3.2 | 30.1 ↑2.2 |

**On-policy sampling and small learning rate mitigate the probability drops in preferred responses.** In the literature, the Direct Preference Optimization (DPO) algorithm is often reported to diminish reasoning capabilities by reducing the likelihood of preferred responses (Yuan et al., 2024; Hong et al., 2024; Meng et al., 2024). In our preliminary experiments, we also observe similar phenomena with a large learning rate (1e-6), where the model's reasoning ability collapses after only a few training steps, preventing convergence to good reasoning performance. In contrast, we find that using on-policy sampling within our online iterative training framework, coupled with a smaller learning rate (2e-7 or 4e-7), the DPO algorithm enhances the model's reasoning abilities. To interpret our observation, we can first write down the gradient of the DPO as follows:

$$\nabla_\theta \mathcal{L}_{DPO}(\pi_\theta, \pi_{\mathrm{ref}}) = -\eta \cdot \sigma\Big(r_\theta(x, y^l) - r_\theta(x, y^w)\Big)\Big[\frac{1}{\pi_\theta(y^w|x)}\nabla_\theta \pi_\theta(y^w|x) - \frac{1}{\pi_\theta(y^l|x)}\nabla_\theta \pi_\theta(y^l|x)\Big],$$

where $r_\theta(x, y) = \eta \log \frac{\pi_\theta(x,y)}{\pi_{\mathrm{ref}}(x,y)}$ is the implicit reward and we use the single-turn one for simplicity. In practice, the probability of the rejected responses typically decrease, and their gradient quickly dominates when $\pi_\theta(y^l|x) << \pi_\theta(y^w|x)$ and the optimization becomes unlearning of the rejected responses. In this case, the probability of the chosen responses cannot increase. This phenomenon was also discussed in the blog Guo et al. (2024a). When we adopt on-policy sampling, it leads to a relatively large probability for both rejected and chosen responses at the initial stage, ensuring that both gradients remain valid and effective. Moreover, a small learning rate prevents the model from deviating too significantly, maintaining the effectiveness of both gradients. We also notice that for the KTO algorithm, the preferred responses and the rejected responses do not appear in pairs. We suspect that the probability of the preferred response increases because the gradients of the rejected response do not dominate in every mini-batch of data. A more comprehensive understanding of the training dynamic of the direct preference learning algorithms remains largely open and we leave a more detailed study of this phenomena to future study.

## D  THEORETICAL PROOFS

### D.1  THEORETICAL RESULTS

In this following, we show that the multi-turn RLHF problem can be solved in a statistically efficient manner under standard assumptions in learning theory literature. In particular, for generality, we target the most challenging scenario with stochastic and unknown transitions, while as aforementioned, multi-turn mathematical reasoning with external tools falls into a relatively easier regime with deterministic transitions. As mentioned in the main paper, we mostly study the KL-regularized target due to the lack of theoretical research on it. The other target of optimizing the rewards has been theoretically studied in Wang et al. (2023b) while the techniques of analyzing mirror-descent-style algorithm and corresponding guarantees have also been developed in Cai et al. (2020), which can be migrated to considering preference feedbacks. Also, to ease the presentation, we consider the scenario with batch size $m = 1$, while the results can be easily generalized to large batches.

First, to measure the online learning process, we define the optimal policy as

$$\pi^* := \arg\max_{\pi} J(\pi) := J(\pi; \mathcal{M}^*, \pi_0), \tag{14}$$

and introduce the standard notion of regret as

$$\text{Reg}(T) := \sum_{t \in [T]} J(\pi^*) - J(\pi_t^1), \tag{15}$$

which represents the cumulative performance loss over $T$ steps comparing the learned policies $[\pi_t^1]_{t=1}^T$ against the optimal policy $\pi^*$. In addition, we consider that a bounded $u^*(x, y) \in [0, B]$ for all $(x, y)$ to maintain a reasonable utillity regime. Also, it is assumed that we have accesses to the following policy improvement oracle, that is analogue to the one considered in Xiong et al. (2024).

**Definition 2** (Policy Improvement Oracle). *For any model $\mathcal{M} = (\mathcal{S}, \mathcal{A}, H, \mathbb{P}, d_0, u)$ and a reference function $\pi_{\text{ref}}$, we can compute the optimal policy associated with the model $[\pi_{\mathcal{M},h}]_{h=1}^H$ iteratively as in equation 4.*

The overall algorithm, i.e., the theoretical version of online iterative M-GSHF, is also summarized in Algorithm 1. At each round $t$, with $\mathcal{D} = \cup_{i=1}^{t-1} \mathcal{D}_i$ as the aggregated dataset, it starts with performing a maximum likelihood estimation (MLE) of the reward function $u^*$ over a set $\mathcal{U}$, whose elements are bounded in $[0, B]$, as

$$\hat{u}_t = \arg\max_{\hat{u} \in \mathcal{U}} L_t(\hat{u})$$
$$:= \sum_{(x, \tau^1, \tau^2, z) \in \cup_{i=1}^{t-1} \mathcal{D}_i} \left[ z \log(\sigma(\hat{u}(\tau^1) - \hat{u}(\tau^2))) + (1 - z) \log(\sigma(\hat{u}(\tau^2) - \hat{u}(\tau^1))) \right], \tag{16}$$

and also an MLE of the transition kernel $\mathbb{P}^*$ over a set $\mathcal{P}$ as

$$\hat{\mathbb{P}}_t = \arg\max_{\hat{\mathbb{P}} \in \mathcal{P}} L_t(\hat{\mathbb{P}}) := \sum_{(\pi, \tau) \in \cup_{i=1}^{t-1} \mathcal{D}_i} \log \hat{\mathbb{P}}^{\pi}(\tau), \tag{17}$$

where $\mathbb{P}^{\pi}(\tau)$ denotes the probability of trajectory $\tau$ under policy $\pi$ and transition kernel $\mathbb{P}$. With the obtained model $\hat{\mathcal{M}}_t = (\hat{u}_t, \hat{\mathbb{P}}_t)$, the Oracle defined in Definition 2 is called with the reference policy $\pi_{\text{ref}}$ set as the initial policy $\pi_0$, whose output is adopted as the main policy $\pi_t^1$.

Then, we specify how to choose a theoretically sound exploration policy $\pi_t^2$. The previous work of Xiong et al. (2024) on single-turn RLHF has demonstrated the intuition that the exploration policy should be in charge of collecting information of the uncertain parts of the environment $\mathcal{M}$, which is thus often selected to maximize one uncertainty measurement. In the multi-turn RLHF setup considered in this work, the following proposition serves as the cornerstone to find a suitable uncertainty measurement to decide the exploration policy. In particular, we can observe that the optimal policy is parameterized by the optimal $Q$-function. If a different set of $Q$-function is adopted for policy parameterization, we can bound its performance as follows.

**Proposition 2** (Value Decomposition Lemma for KL-regularized MDP). *If considering a set of $Q$-functions $[\hat{Q}_h]_{h=1}^H$ and a reference policy $\pi_{\text{ref}}$ with the induced policy $\hat{\pi}$ as*

$$\hat{\pi}_h(a_h|s_h) \propto \pi_{\text{ref},h}(a_h|s_h) \cdot \exp\left(\hat{Q}_h(s_h, a_h)/\eta\right),$$

*and the corresponding set of $V$-functions $[\hat{V}_h]_{h=1}^H$ as*

$$\hat{V}_h(s_h) = \mathbb{E}_{a_h \sim \hat{\pi}_h(\cdot|s_h)} \left[ \hat{Q}_h(s_h, a_h) \right] - \eta D_{\mathrm{KL}}(\hat{\pi}_h(\cdot|s_h), \pi_{\mathrm{ref},h}(\cdot|s_h)), \qquad \hat{V}_{H+1}(s_{H+1}) = 0,$$

*for any comparator policy $\pi$, it holds that*

$$
\begin{aligned}
&J(\pi) - J(\hat{\pi}) \\
&= \mathbb{E}_{d_0,\pi,\mathbb{P}^*}[u^*(s_H, a_H)] - \mathbb{E}_{d_0,\hat{\pi},\mathbb{P}^*}[u^*(s_H, a_H)] \\
&\quad + \sum_{h \in [H]} \mathbb{E}_{d_0,\pi,\mathbb{P}^*} \left[ \hat{V}_{H+1}(s_{h+1}) - \hat{Q}_h(s_h, a_h) \right] - \sum_{h \in [H]} \mathbb{E}_{d_0,\hat{\pi},\mathbb{P}^*} \left[ \hat{V}_{H+1}(s_{h+1}) - \hat{Q}_h(s_h, a_h) \right] \\
&\quad - \eta \cdot \sum_{h \in [H]} \mathbb{E}_{d_0,\pi,\mathbb{P}^*} \left[ D_{\mathrm{KL}}(\pi_h(\cdot|s_h), \hat{\pi}_h(\cdot|s_h)) \right],
\end{aligned}
$$

*where the expectation $\mathbb{E}_{d_0,\pi,\mathbb{P}^*}$ is with respect to the prompt and response (i.e., the trajectory) generated following $d_0, \mathbb{P}^*$ and $\pi$.*

Based on Proposition 2, the exploration policy $\pi_t^2$ is selected as

$$
\begin{aligned}
\pi_t^2 = \arg\max_\pi \max_{\widetilde{u} \in \widetilde{\mathcal{U}}_t, \widetilde{\mathbb{P}} \in \widetilde{\mathcal{P}}_t} & \; \mathbb{E}_{d_0,\pi,\widetilde{\mathbb{P}}}[\widetilde{u}(s_H, a_H)] - \mathbb{E}_{d_0,\pi_t^1,\widetilde{\mathbb{P}}}[\widetilde{u}(s_H, a_H)] \\
& - \left( \mathbb{E}_{d_0,\pi,\widetilde{\mathbb{P}}}[\hat{u}_t(s_H, a_H)] - \mathbb{E}_{d_0,\pi_t^1,\widetilde{\mathbb{P}}}[\hat{u}_t(s_H, a_H)] \right) \\
& + \sum_{h \in [H]} \mathbb{E}_{d_0,\pi,\widetilde{\mathbb{P}}} \left[ \hat{V}_{t,h+1}(s_{h+1}) - \left[ \hat{\mathbb{P}}_{t,h} \hat{V}_{t,h+1} \right] (s_h, a_h) \right],
\end{aligned}
\tag{18}
$$

where $\widetilde{\mathcal{U}}_t$ and $\widetilde{\mathcal{P}}_t$ are two confidence sets defined as

$$
\begin{aligned}
\widetilde{\mathcal{U}}_t &= \{ u \in \mathcal{U} : L_t(u) \geq L_t(\hat{u}_t) - c_1 \log(|\mathcal{U}|T/\delta) \}, \\
\widetilde{\mathcal{P}}_t &= \{ \mathbb{P} \in \mathcal{P} : L_t(\mathbb{P}) \geq L_t(\hat{\mathbb{P}}_t) - c_1 \log(|\mathcal{P}|T/\delta) \}
\end{aligned}
\tag{19}
$$

with $c_1$ denoting an absolute constant here. Note that for the theoretical convenience, we have assumed $\mathcal{U}$ and $\mathcal{P}$ are finite here, which can be extended to the infinite case using standard discretization techniques. It can be observed that $\pi_t^2$ is selected to maximize a combination of uncertainty from estimations of both rewards and transitions. If considering known transitions (i.e., without the need to estimate $\mathbb{P}$), the uncertainty from the estimation of transitions diminishes, which leads to a similar uncertainty measurement adopted in Xiong et al. (2024).

The following theorem establishes a rigorous guarantee for the regret incurred.

**Theorem 2.** *Assuming $u^* \in \mathcal{U}$ and $\mathbb{P}^* \in \mathcal{P}$, with probability at least $1 - \delta$, we have that*

$$
\begin{aligned}
\mathrm{Reg}(T) \lesssim &\kappa^{-1} B \sqrt{d_{\mathcal{U}} T \log(|\mathcal{U}|T/\delta)} + B^2 H \xi(d_{\mathcal{P}}, T, c_2 \log(|\mathcal{P}|HT/\delta)) \\
& - \eta \cdot \sum_{t \in [T]} \sum_{h \in [H]} \mathbb{E}_{d_0,\pi^*,\mathbb{P}^*} \left[ D_{\mathrm{KL}}(\pi_h^*(\cdot|s_h), \pi_{t,h}^1(\cdot|s_h)) \right],
\end{aligned}
$$

*where $\kappa := 1/(2 + \exp(-B) + \exp(B))$, $c_2$ is an absolute constant, $d_{\mathcal{U}}$ is the Eluder coefficient defined in Definition 3 while $d_{\mathcal{P}}$ and $\xi(\cdot)$ are from the generalized Eluder-type condition defined in Definition 4.*

We note that the Eluder coefficient and the generalized Eluder-type condition are standard and well-adopted conditions in the theoretical studies on RL (Zhang, 2023; Zhong et al., 2022; Liu et al., 2023a; Xie et al., 2022; Agarwal et al., 2023) and also RLHF (Zhan et al., 2023; Wang et al., 2023b; Ye et al., 2024). Moreover, for a board class of RL problems (see Zhang (2023); Liu et al. (2023a) for more details), the Eluder coefficient $d_{\mathcal{U}}$ is small and the condition is satisfied with $\xi(d_{\mathcal{P}}, T, c_2 \log(|\mathcal{P}|HT/\delta)) \lesssim \sqrt{d_{\mathcal{P}} T \log(|\mathcal{P}|HT/\delta)}$, which implies that the regret of theoretical version of Algorithm 1 is sublinear in $T$, further evidencing its statistical efficiency.

### D.2 PROOF OF PROPOSITION 2

*Proof of Proposition 2.* For one policy $\pi$, starting with $V^\pi_{\mathcal{M},H+1} = 0$, we recursively define its $V$-value and $Q$-value functions on one model $\mathcal{M} = (\mathcal{S}, \mathcal{A}, H, \mathbb{P}, d_0, u)$ and the reference policy $\pi_{\text{ref}}$ as

$$Q^\pi_{\mathcal{M},h}(s_h, a_h) := \begin{cases} u(s_H, a_H), & \text{if } h = H, \\ \mathbb{E}_{o_h \sim \mathbb{P}_h(\cdot|s_h,a_h)}[V^\pi_{\mathcal{M},h+1}(s_{h+1})], & \text{if } h \le H-1, \end{cases}$$

$$V^\pi_{\mathcal{M},h}(s_h) := \mathbb{E}_{a_h \sim \pi_h(\cdot|s_h)}\left[Q^\pi_{\mathcal{M},h}(s_h,a_h) - \eta \cdot D_{\text{KL}}\big(\pi_h(\cdot|s_h), \pi_{\text{ref},h}(\cdot|s_h)\big)\right].$$

It is noted that with the optimal policy $\pi_\mathcal{M}$, $Q_{\mathcal{M},h} = Q^{\pi_\mathcal{M}}_{\mathcal{M},h}$ and $V_{\mathcal{M},h} = V^{\pi_\mathcal{M}}_{\mathcal{M},h}$. In the following discussions, we exclusively focus on the model $\mathcal{M}^* = (\mathcal{S}, \mathcal{A}, H, \mathbb{P}^*, d_0, u^*)$ with abbreviations $Q^\pi_h = Q^\pi_{\mathcal{M}^*,h}$ and $V^\pi_h = V^\pi_{\mathcal{M}^*,h}$.

For any comparator policy $\pi$, it holds that

$$J(\pi) - J(\hat\pi) = \mathbb{E}_{d_0}\left[V^\pi_1(s_1) - \hat V_1(s_1)\right] - \mathbb{E}_{d_0}\left[V^{\hat\pi}_1(s_1) - \hat V_1(s_1)\right],$$

For any $h \in [H]$, we can obtain that

$$\mathbb{E}_{d_0,\pi_{1:h-1},\mathbb{P}^*_{1:h-1}}\left[V^\pi_h(s_h) - \hat V_h(s_h)\right] - \mathbb{E}_{d_0,\hat\pi_{1:h-1},\mathbb{P}^*_{1:h-1}}\left[V^{\hat\pi}_h(s_h) - \hat V_h(s_h)\right]$$

$$\overset{(a)}{=} \mathbb{E}_{d_0,\pi_{1:h-1},\mathbb{P}^*_{1:h-1}}\left[\mathbb{E}_{\pi_h}\left[Q^\pi_h(s_h,a_h)\right] - \eta D_{\text{KL}}\left(\pi_h(\cdot|s_h), \pi_{\text{ref},h}(\cdot|s_h)\right)\right]$$

$$- \mathbb{E}_{d_0,\pi_{1:h-1},\mathbb{P}^*_{1:h-1}}\left[\mathbb{E}_{\hat\pi_h}\left[\hat Q_h(s_h,a_h)\right] - \eta D_{\text{KL}}\left(\hat\pi_h(\cdot|s_h), \pi_{\text{ref},h}(\cdot|s_h)\right)\right]$$

$$- \mathbb{E}_{d_0,\hat\pi_{1:h-1},\mathbb{P}^*_{1:h-1}}\left[\mathbb{E}_{\hat\pi_h}\left[Q^{\hat\pi}_h(s_h,a_h)\right] - \eta D_{\text{KL}}(\hat\pi_h(\cdot|s_h), \pi_{\text{ref},h}(\cdot|s_h))\right]$$

$$+ \mathbb{E}_{d_0,\hat\pi_{1:h-1},\mathbb{P}^*_{1:h-1}}\left[\mathbb{E}_{\hat\pi_h}\left[\hat Q_h(s_h,a_h)\right] - \eta D_{\text{KL}}(\hat\pi_h(\cdot|s_h), \pi_{\text{ref},h}(\cdot|s_h))\right]$$

$$= \mathbb{E}_{d_0,\pi_{1:h},\mathbb{P}^*_{1:h-1}}\left[Q^\pi_h(s_h,a_h) - \hat Q_h(s_h,a_h)\right] - \mathbb{E}_{d_0,\hat\pi_{1:h},\mathbb{P}^*_{1:h-1}}\left[Q^{\hat\pi}_h(s_h,a_h) - \hat Q_h(s_h,a_h)\right]$$

$$+ \mathbb{E}_{d_0,\pi_{1:h-1},\mathbb{P}^*_{1:h-1}}\underbrace{\left[\mathbb{E}_{\pi_h}\left[\hat Q_h(s_h,a_h)\right] - \mathbb{E}_{\hat\pi_h}\left[\hat Q_h(s_h,a_h)\right]\right]}_{\text{term (I)}}$$

$$- \eta \cdot \mathbb{E}_{d_0,\pi_{1:h-1},\mathbb{P}^*_{1:h-1}}\left[D_{\text{KL}}\left(\pi_h(\cdot|s_h), \pi_{\text{ref},h}(\cdot|s_h)\right)\right]$$

$$+ \eta \cdot \mathbb{E}_{d_0,\pi_{1:h-1},\mathbb{P}^*_{1:h-1}}\left[D_{\text{KL}}\left(\hat\pi_h(\cdot|s_h), \pi_{\text{ref},h}(\cdot|s_h)\right)\right]$$

$$\overset{(b)}{=} \mathbb{E}_{d_0,\pi_{1:h},\mathbb{P}^*_{1:h-1}}\left[Q^\pi_h(s_h,a_h) - \hat Q_h(s_h,a_h)\right] - \mathbb{E}_{d_0,\hat\pi_{1:h},\mathbb{P}^*_{1:h-1}}\left[Q^{\hat\pi}_h(s_h,a_h) - \hat Q_h(s_h,a_h)\right]$$

$$- \eta \cdot \mathbb{E}_{d_0,\pi_{1:h-1},\mathbb{P}^*_{1:h-1}}\left[D_{\text{KL}}\left(\pi_h(\cdot|s_h), \hat\pi_h(\cdot|s_h)\right)\right].$$

In the above derivation, equation (a) is from the definitions of $Q^\pi$ and $V^\pi$, and the relationship between $\hat Q$ and $\hat V$. The equation (b) is because

$$\text{(term I)} := \mathbb{E}_{\pi_h}\left[\hat Q_h(s_h,a_h)\right] - \mathbb{E}_{\hat\pi_h}\left[\hat Q_h(s_h,a_h)\right]$$

$$= \eta \cdot \mathbb{E}_{\pi_h}\left[\log \frac{\hat\pi_h(a_h|s_h)}{\pi_{\text{ref},h}(a_h|s_h)}\right] - \eta \cdot \mathbb{E}_{\hat\pi_h}\left[\log \frac{\hat\pi_h(a_h|s_h)}{\pi_{\text{ref},h}(a_h|s_h)}\right]$$

$$= \eta \cdot D_{\text{KL}}\left(\pi_h(\cdot|s_h), \pi_{\text{ref},h}(\cdot|s_h)\right) - \eta \cdot D_{\text{KL}}\left(\pi_h(\cdot|s_h), \hat\pi_h(\cdot|s_h)\right)$$

$$- \eta \cdot D_{\text{KL}}\left(\hat\pi_h(\cdot|s_h), \pi_{\text{ref},h}(\cdot|s_h)\right).$$

where the second equation is from the relationship that

$$\hat Q_h(s_h,a_h) = \eta \cdot \log \frac{\hat\pi_h(a_h|s_h)}{\pi_{\text{ref},h}(a_h|s_h)} - \eta \cdot \log \hat Z_h(s_h).$$

Furthermore, if $h = H$, we can obtain that

$$\mathbb{E}_{d_0,\pi_{1:H-1},\mathbb{P}^*_{1:H-1}}\left[V^\pi_H(s_H) - \hat V_H(s_H)\right] - \mathbb{E}_{d_0,\hat\pi_{1:H-1},\mathbb{P}^*_{1:H-1}}\left[V^{\hat\pi}_H(s_H) - \hat V_H(s_H)\right]$$

$$= \mathbb{E}_{d_0, \pi_{1:H}, \mathbb{P}^*_{1:H-1}} \left[ u^*(s_H, a_H) - \hat{Q}_H(s_H, a_H) \right] - \mathbb{E}_{d_0, \hat{\pi}_{1:H}, \mathbb{P}^*_{1:H-1}} \left[ u^*(s_H, a_H) - \hat{Q}_H(s_H, a_H) \right]$$
$$- \eta \cdot \mathbb{E}_{d_0, \pi_{1:H-1}, \mathbb{P}^*_{1:H-1}} \left[ D_{\mathrm{KL}} \left( \pi_H(\cdot|s_H), \hat{\pi}_H(\cdot|s_H) \right) \right]$$
$$= \mathbb{E}_{d_0, \pi_{1:H}, \mathbb{P}^*_{1:H-1}} \left[ u^*(s_H, a_H) \right] - \mathbb{E}_{d_0, \hat{\pi}_{1:H}, \mathbb{P}^*_{1:H-1}} \left[ u^*(s_H, a_H) \right]$$
$$+ \mathbb{E}_{d_0, \pi_{1:H}, \mathbb{P}^*_{1:H}} \left[ \hat{V}_{H+1}(s_{H+1}) - \hat{Q}_H(s_H, a_H) \right] - \mathbb{E}_{d_0, \hat{\pi}_{1:H}, \mathbb{P}^*_{1:H}} \left[ \hat{V}_{H+1}(s_{H+1}) - \hat{Q}_H(s_H, a_H) \right]$$
$$- \eta \cdot \mathbb{E}_{d_0, \pi_{1:H-1}, \mathbb{P}^*_{1:H-1}} \left[ D_{\mathrm{KL}} \left( \pi_H(\cdot|s_H) || \hat{\pi}_H(\cdot|s_H) \right) \right],$$

where the second equality leverages that $\hat{V}_{H+1}(s_{H+1}) = 0$; otherwise, for all $h \leq H - 1$, it holds that

$$\mathbb{E}_{d_0, \pi_{1:h-1}, \mathbb{P}^*_{1:h-1}} \left[ V_h^{\pi}(s_h) - \hat{V}_h(s_h) \right] - \mathbb{E}_{d_0, \hat{\pi}_{1:h-1}, \mathbb{P}^*_{1:h-1}} \left[ V_h^{\hat{\pi}}(s_h) - \hat{V}_h(s_h) \right]$$
$$= \mathbb{E}_{d_0, \pi_{1:h}, \mathbb{P}^*_{1:h-1}} \left[ Q_h^{\pi}(s_h, a_h) - \hat{Q}_h(s_h, a_h) \right] - \mathbb{E}_{d_0, \hat{\pi}_{1:h}, \mathbb{P}^*_{1:h-1}} \left[ Q_h^{\hat{\pi}}(s_h, a_h) - \hat{Q}_h(s_h, a_h) \right]$$
$$- \eta \cdot \mathbb{E}_{d_0, \pi_{1:h-1}, \mathbb{P}^*_{1:h-1}} \left[ D_{\mathrm{KL}} \left( \pi_h(\cdot|s_h) || \hat{\pi}_h(\cdot|s_h) \right) \right]$$
$$= \mathbb{E}_{d_0, \pi_{1:h}, \mathbb{P}^*_{1:h}} \left[ \hat{V}_{h+1}(s_{h+1}) - \hat{Q}_h(s_h, a_h) \right] - \mathbb{E}_{d_0, \hat{\pi}_{1:h}, \mathbb{P}^*_{1:h}} \left[ \hat{V}_{h+1}(s_{h+1}) - \hat{Q}_h(s_h, a_h) \right]$$
$$- \eta \cdot \mathbb{E}_{d_0, \pi_{1:h-1}, \mathbb{P}^*_{1:h-1}} \left[ D_{\mathrm{KL}} \left( \pi_h(\cdot|s_h) || \hat{\pi}_h(\cdot|s_h) \right) \right]$$
$$+ \mathbb{E}_{d_0, \pi_{1:h}, \mathbb{P}^*_{1:h}} \left[ V_{h+1}^{\pi}(s_{h+1}) - \hat{V}_{h+1}(s_{h+1}) \right] - \mathbb{E}_{d_0, \pi_{1:h}, \mathbb{P}^*_{1:h}} \left[ V_{h+1}^{\hat{\pi}}(s_{h+1}) - \hat{V}_{h+1}(s_{h+1}) \right].$$

The proposition can be obtained by iteratively using the above relationship for $h \in [H]$. $\qquad\square$

### D.3 Proof of Theorem 2

First, with the assumption $u^* \in \mathcal{U}$ and $\mathbb{P}^* \in \mathcal{P}$, the following lemma demonstrates that $\widetilde{\mathcal{U}}_t$ and $\widetilde{\mathcal{P}}_t$ are valid confidence sets.

**Lemma 1** (Proposition B.1 from Liu et al. (2023a))**.** *There exists an absolute constant $c_1$ such that for any $\delta \in (0, 1]$, with probability at least $1 - \delta$, for all $t \in [T]$, $\hat{u} \in \mathcal{U}$, and $\hat{\mathbb{P}} \in \mathcal{P}$, it holds that*

$$L_t(\hat{u}) - L_t(u^*) \leq c_1 \log(|\mathcal{U}|T/\delta), \qquad L_t(\hat{\mathbb{P}}) - L_t(\mathbb{P}^*) \leq c_1 \log(|\mathcal{P}|T/\delta),$$

*which implies that $u^* \in \widetilde{\mathcal{U}}_t$ and $\mathbb{P}^* \in \widetilde{\mathcal{P}}_t$.*

Then, we provide an additional lemma demonstrating the in-sample error of the MLE and optimistic estimators.

**Lemma 2.** *There exists an absolute constant $c_2$ such that for any $\delta \in (0, 1]$, with probability at least $1 - \delta$, for all $t \in [T]$, we have*

$$\sum_{i<t} \left| \sigma \left( \hat{u}_t(s_{i,H}^2, a_{i,H}^2) - \hat{u}_t(s_{i,H}^1, a_{i,H}^1) \right) - \sigma \left( u^*(s_{i,H}^2, a_{i,H}^2) - u^*(s_{i,H}^1, a_{i,H}^1) \right) \right|^2 \leq c_2 \log(|\mathcal{U}|T/\delta);$$

$$\sum_{i<t} \left| \sigma \left( \widetilde{u}_t(s_{i,H}^2, a_{i,H}^2) - \widetilde{u}_t(s_{i,H}^1, a_{i,H}^1) \right) - \sigma \left( u^*(s_{i,H}^2, a_{i,H}^2) - u^*(s_{i,H}^1, a_{i,H}^1) \right) \right|^2 \leq c_2 \log(|\mathcal{U}|T/\delta),$$

*and for all $t \in [T]$, $h \in [H]$, we have*

$$\sum_{j \in \{1,2\}} \sum_{h \in [H]} \sum_{i<t} \mathrm{TV} \left( \{d_0, \pi_i^j, [\mathbb{P}^*_{1:h-1}, \hat{\mathbb{P}}_{t,h}, \mathbb{P}^*_{h+1:H}]\}, \{d_0, \pi_i^j, \mathbb{P}^*_{1:H}\} \right)^2 \leq c_2 \log(|\mathcal{P}|HT/\delta);$$

$$\sum_{j \in \{1,2\}} \sum_{h \in [H]} \sum_{i<t} \mathrm{TV} \left( \{d_0, \pi_i^j, [\mathbb{P}^*_{1:h-1}, \widetilde{\mathbb{P}}_{t,h}, \mathbb{P}^*_{h+1:H}]\}, \{d_0, \pi_i^j, \mathbb{P}^*_{1:H}\} \right)^2 \leq c_2 \log(|\mathcal{P}|HT/\delta),$$

*where $\mathrm{TV}(\{d_0, \pi, \mathbb{P}\}, \{d_0, \pi', \mathbb{P}'\})$ denotes the TV distance between the probability distributions over the trajectories induced by $d_0, \pi, \mathbb{P}$ and $d_0, \pi', \mathbb{P}'$.*

*Proof of Lemma 2.* First, for $\widetilde{u}_t$, we can obtain that with probability at least $1 - \delta$, there exists an absolute constant $c$ such that for all $t \in [T]$,

$$\sum_{i<t} \left| \sigma \left( \widetilde{u}_t(s_{i,H}^2, a_{i,H}^2) - \widetilde{u}_t(s_{i,H}^1, a_{i,H}^1) \right) - \sigma \left( u^*(s_{i,H}^2, a_{i,H}^2) - u^*(s_{i,H}^1, a_{i,H}^1) \right) \right|^2$$

$$
\begin{aligned}
&\leq c \cdot \sum_{i<t} \log \frac{z_i \cdot \sigma\left(u^*(s_{i,H}^1, a_{i,H}^1) - u^*(s_{i,H}^2, a_{i,H}^2)\right) + (1 - z_i) \cdot \sigma\left(u^*(s_{i,H}^2, a_{i,H}^2) - u^*(s_{i,H}^1, a_{i,H}^1)\right)}{z_i \cdot \sigma\left(\widetilde{u}_t(s_{i,H}^1, a_{i,H}^1) - \widetilde{u}_t(s_{i,H}^2, a_{i,H}^2)\right) + (1 - z_i) \cdot \sigma\left(\widetilde{u}_t(s_{i,H}^2, a_{i,H}^2) - \widetilde{u}_t(s_{i,H}^1, a_{i,H}^1)\right)} \\
&\quad + c \cdot \log(|\mathcal{U}| T / \delta) \\
&= c\left(L_t(u^*) - L_t(\widetilde{u}_t) + \log(|\mathcal{U}| T / \delta)\right) \\
&\leq c\left(L_t(u^*) - L_t(\hat{u}_t) + c_1 \log(|\mathcal{U}| T / \delta) + \log(|\mathcal{U}| T / \delta)\right) \\
&\leq c_2 \log(|\mathcal{U}| T / \delta).
\end{aligned}
$$

where the first inequality is from Proposition B.2 from Liu et al. (2023a) and the second inequality uses Lemma 1. The result for $\hat{u}_t$ can be similarly established.

Then, following similar steps, for $\widetilde{\mathbb{P}}_t$, we can obtain that with probability at least $1 - \delta$, there exists an absolute constant $c$ such that for all $t \in [T]$,

$$
\begin{aligned}
&\sum_{j \in \{1,2\}} \sum_{h \in [H]} \sum_{i<t} \mathrm{TV}\left(\{d_0, \pi_i^j, [\mathbb{P}_{1:h-1}^*, \widetilde{\mathbb{P}}_{t,h}, \mathbb{P}_{h+1:H}^*]\}, \{d_0, \pi_i^j, \mathbb{P}_{1:H}^*\}\right)^2 \\
&\leq \sum_{j \in \{1,2\}} \sum_{h \in [H]} c \cdot \left(\sum_{i<t} \log \frac{\mathbb{P}_h^*(s_{i,h+1}^j | s_{i,h}^j, a_{i,h}^j)}{\widetilde{\mathbb{P}}_{t,h}(s_{i,h+1}^j | s_{i,h}^j, a_{i,h}^j)} + \log(|\mathcal{P}_h| HT / \delta)\right) \\
&= c \cdot \left(\sum_{j \in \{1,2\}} \sum_{i<t} \log \frac{\mathbb{P}^{*,\pi_i^j}(\tau_i^j)}{\widetilde{\mathbb{P}}_t^{\pi_i^j}(\tau_i^j)} + 2\log(|\mathcal{P}| HT / \delta)\right) \\
&= c \cdot \left(L_t(\mathbb{P}^*) - L_t(\widetilde{\mathbb{P}}_t) + 2\log(|\mathcal{P}| HT / \delta)\right) \\
&\leq c \cdot \left(L_t(\mathbb{P}^*) - L_t(\hat{\mathbb{P}}_t) + c_1 \log(|\mathcal{P}| T / \delta) + 2\log(|\mathcal{P}| HT / \delta)\right) \\
&\leq c_2 \log(|\mathcal{P}| HT / \delta).
\end{aligned}
$$

The result for $\hat{\mathbb{P}}_t$ can also be similarly established. $\qquad\square$

*Proof of Theorem 2.* In the following proofs, we omit the KL term in the decomposition to ease the presentation. Then, with probability at least $1 - \delta$, for all $t \in [T]$, we can obtain that

$$
\begin{aligned}
&J(\pi^*) - J(\pi_t^1) \\
&= \mathbb{E}_{d_0, \pi^*, \mathbb{P}^*}\left[u^*(s_H, a_H)\right] - \mathbb{E}_{d_0, \pi_t^1, \mathbb{P}^*}\left[u^*(s_H, a_H)\right] - \left(\mathbb{E}_{d_0, \pi^*, \mathbb{P}^*}\left[\hat{u}_t(s_H, a_H)\right] - \mathbb{E}_{d_0, \pi_t^1, \mathbb{P}^*}\left[\hat{u}_t(s_H, a_H)\right]\right) \\
&\quad + \sum_{h \in [H]} \mathbb{E}_{d_0, \pi^*, \mathbb{P}^*}\left[\hat{V}_{t,h+1}(s_{h+1}) - \left[\hat{\mathbb{P}}_{t,h} \hat{V}_{t,h+1}\right](s_h, a_h)\right] - \sum_{h \in [H]} \mathbb{E}_{d_0, \pi_t^1, \mathbb{P}^*}\left[\hat{V}_{t,h+1}(s_{h+1}) - \left[\hat{\mathbb{P}}_{t,h} \hat{V}_{t,h+1}\right](s_h, a_h)\right] \\
&\leq \underbrace{\mathbb{E}_{d_0, \pi_t^2, \widetilde{\mathbb{P}}_t}\left[\widetilde{u}_t(s_H, a_H)\right] - \mathbb{E}_{d_0, \pi_t^1, \widetilde{\mathbb{P}}_t}\left[\widetilde{u}_t(s_H, a_H)\right] - \left(\mathbb{E}_{d_0, \pi_t^2, \widetilde{\mathbb{P}}_t}\left[\hat{u}_t(s_H, a_H)\right] - \mathbb{E}_{d_0, \pi_t^1, \widetilde{\mathbb{P}}_t}\left[\hat{u}_t(s_H, a_H)\right]\right)}_{\text{term (I)}_t} \\
&\quad + \underbrace{\sum_{h \in [H]} \mathbb{E}_{d_0, \pi_t^2, \widetilde{\mathbb{P}}_t}\left[\hat{V}_{t,h+1}(s_{h+1}) - \left[\hat{\mathbb{P}}_{t,h} \hat{V}_{t,h+1}\right](s_h, a_h)\right] + \sum_{h \in [H]} \mathbb{E}_{d_0, \pi_t^1, \mathbb{P}^*}\left[\left[\hat{\mathbb{P}}_{t,h} \hat{V}_{t,h+1}\right](s_h, a_h) - \hat{V}_{t,h+1}(s_{h+1})\right]}_{\text{term (II)}_t},
\end{aligned}
$$

where the inequality is from the definition of $\pi_t^2$ and the fact that $(u^*, \mathbb{P}^*) \in \widetilde{\mathcal{U}}_t \times \widetilde{\mathcal{P}}_t$ from Lemma 1.

We define the following terms:

$$
\text{term (A)}_t := \mathbb{E}_{d_0, \pi_t^2, \mathbb{P}^*}\left[\widetilde{u}_t(s_H, a_H)\right] - \mathbb{E}_{d_0, \pi_t^1, \mathbb{P}^*}\left[\widetilde{u}_t(s_H, a_H)\right] - \left(\mathbb{E}_{d_0, \pi_t^2, \mathbb{P}^*}\left[u^*(s_H, a_H)\right] - \mathbb{E}_{d_0, \pi_t^1, \mathbb{P}^*}\left[u^*(s_H, a_H)\right]\right),
$$

$$
\text{term (B)}_t := \mathbb{E}_{d_0, \pi_t^2, \mathbb{P}^*}\left[u^*(s_H, a_H)\right] - \mathbb{E}_{d_0, \pi_t^1, \mathbb{P}^*}\left[u^*(s_H, a_H)\right] - \left(\mathbb{E}_{d_0, \pi_t^2, \mathbb{P}^*}\left[\hat{u}_t(s_H, a_H)\right] - \mathbb{E}_{d_0, \pi_t^1, \mathbb{P}^*}\left[\hat{u}_t(s_H, a_H)\right]\right),
$$

$$
\text{term (C)}_t := \sum_{j \in \{1,2\}} \sum_{h \in [H]} \mathbb{E}_{d_0, \pi_t^j, \mathbb{P}^*}\left[\mathrm{TV}\left(\widetilde{\mathbb{P}}_{t,h}(\cdot | s_h, a_h), \mathbb{P}_h^*(\cdot | s_h, a_h)\right)\right],
$$

$$
\text{term (D)}_t := \sum_{j \in \{1,2\}} \sum_{h \in [H]} \mathbb{E}_{d_0, \pi_t^j, \mathbb{P}^*}\left[\mathrm{TV}\left(\hat{\mathbb{P}}_{t,h}(\cdot | s_h, a_h), \mathbb{P}_h^*(\cdot | s_h, a_h)\right)\right].
$$

For term $(\text{I})_t$, we have that

$$\text{term (I)}_t := \mathbb{E}_{d_0,\pi_t^2,\widetilde{\mathbb{P}}_t}\left[\widetilde{u}_t(s_H,a_H)\right] - \mathbb{E}_{d_0,\pi_t^1,\widetilde{\mathbb{P}}_t}\left[\widetilde{u}_t(s_H,a_H)\right] - \left(\mathbb{E}_{d_0,\pi_t^2,\widetilde{\mathbb{P}}_t}\left[\hat{u}_t(s_H,a_H)\right] - \mathbb{E}_{d_0,\pi_t^1,\widetilde{\mathbb{P}}_t}\left[\hat{u}_t(s_H,a_H)\right]\right)$$

$$= \mathbb{E}_{d_0,\pi_t^2,\mathbb{P}^*}\left[\widetilde{u}_t(s_H,a_H)\right] - \mathbb{E}_{d_0,\pi_t^1,\mathbb{P}^*}\left[\widetilde{u}_t(s_H,a_H)\right] - \left(\mathbb{E}_{d_0,\pi_t^2,\mathbb{P}^*}\left[u_t^*(s_H,a_H)\right] - \mathbb{E}_{d_0,\pi_t^1,\mathbb{P}^*}\left[u_t^*(s_H,a_H)\right]\right)$$

$$+ \mathbb{E}_{d_0,\pi_t^2,\mathbb{P}^*}\left[u_t^*(s_H,a_H)\right] - \mathbb{E}_{d_0,\pi_t^1,\mathbb{P}^*}\left[u_t^*(s_H,a_H)\right] - \left(\mathbb{E}_{d_0,\pi_t^2,\mathbb{P}^*}\left[\hat{u}_t(s_H,a_H)\right] - \mathbb{E}_{d_0,\pi_t^1,\mathbb{P}^*}\left[\hat{u}_t(s_H,a_H)\right]\right)$$

$$+ \mathbb{E}_{d_0,\pi_t^2,\widetilde{\mathbb{P}}_t}\left[\widetilde{u}_t(s_H,a_H)\right] - \mathbb{E}_{d_0,\pi_t^1,\widetilde{\mathbb{P}}_t}\left[\widetilde{u}_t(s_H,a_H)\right] - \left(\mathbb{E}_{d_0,\pi_t^2,\mathbb{P}^*}\left[\widetilde{u}_t(s_H,a_H)\right] - \mathbb{E}_{d_0,\pi_t^1,\mathbb{P}^*}\left[\widetilde{u}_t(s_H,a_H)\right]\right)$$

$$+ \mathbb{E}_{d_0,\pi_t^2,\mathbb{P}^*}\left[\hat{u}_t(s_H,a_H)\right] - \mathbb{E}_{d_0,\pi_t^1,\mathbb{P}^*}\left[\hat{u}_t(s_H,a_H)\right] - \left(\mathbb{E}_{d_0,\pi_t^2,\widetilde{\mathbb{P}}_t}\left[\hat{u}_t(s_H,a_H)\right] - \mathbb{E}_{d_0,\pi_t^1,\widetilde{\mathbb{P}}_t}\left[\hat{u}_t(s_H,a_H)\right]\right)$$

$$\leq \mathbb{E}_{d_0,\pi_t^2,\mathbb{P}^*}\left[\widetilde{u}_t(s_H,a_H)\right] - \mathbb{E}_{d_0,\pi_t^1,\mathbb{P}^*}\left[\widetilde{u}_t(s_H,a_H)\right] - \left(\mathbb{E}_{d_0,\pi_t^2,\mathbb{P}^*}\left[u_t^*(s_H,a_H)\right] - \mathbb{E}_{d_0,\pi_t^1,\mathbb{P}^*}\left[u_t^*(s_H,a_H)\right]\right)$$

$$+ \mathbb{E}_{d_0,\pi_t^2,\mathbb{P}^*}\left[u_t^*(s_H,a_H)\right] - \mathbb{E}_{d_0,\pi_t^1,\mathbb{P}^*}\left[u_t^*(s_H,a_H)\right] - \left(\mathbb{E}_{d_0,\pi_t^2,\mathbb{P}^*}\left[\hat{u}_t(s_H,a_H)\right] - \mathbb{E}_{d_0,\pi_t^1,\mathbb{P}^*}\left[\hat{u}_t(s_H,a_H)\right]\right)$$

$$+ 4B \cdot \text{TV}\left(\{d_0,\pi_t^1,\widetilde{\mathbb{P}}_t\},\{d_0,\pi_t^1,\mathbb{P}^*\}\right) + 4B \cdot \text{TV}\left(\{d_0,\pi_t^2,\widetilde{\mathbb{P}}_t\},\{d_0,\pi_t^2,\mathbb{P}\}\right)$$

$$\leq \underbrace{\mathbb{E}_{d_0,\pi_t^2,\mathbb{P}^*}\left[\widetilde{u}_t(s_H,a_H)\right] - \mathbb{E}_{d_0,\pi_t^1,\mathbb{P}^*}\left[\widetilde{u}_t(s_H,a_H)\right] - \left(\mathbb{E}_{d_0,\pi_t^2,\mathbb{P}^*}\left[u_t^*(s_H,a_H)\right] - \mathbb{E}_{d_0,\pi_t^1,\mathbb{P}^*}\left[u_t^*(s_H,a_H)\right]\right)}_{\text{term (A)}_t}$$

$$+ \underbrace{\mathbb{E}_{d_0,\pi_t^2,\mathbb{P}^*}\left[u_t^*(s_H,a_H)\right] - \mathbb{E}_{d_0,\pi_t^1,\mathbb{P}^*}\left[u_t^*(s_H,a_H)\right] - \left(\mathbb{E}_{d_0,\pi_t^2,\mathbb{P}^*}\left[\hat{u}_t(s_H,a_H)\right] - \mathbb{E}_{d_0,\pi_t^1,\mathbb{P}^*}\left[\hat{u}_t(s_H,a_H)\right]\right)}_{\text{term (B)}_t}$$

$$+ 4B \cdot \underbrace{\sum_{j\in\{1,2\}}\sum_{h\in[H]}\mathbb{E}_{d_0}\mathbb{E}_{\pi_t^j,\mathbb{P}^*}\left[\text{TV}\left(\widetilde{\mathbb{P}}_{t,h}(\cdot|s_h,a_h),\mathbb{P}_h^*(\cdot|s_h,a_h)\right)\right]}_{\text{term (C)}_t}.$$

For term $(\text{II})_t$, we have that

$$\text{term (II)}_t = \sum_{h\in[H]}\mathbb{E}_{d_0,\pi_t^2,\widetilde{\mathbb{P}}_t}\left[\hat{V}_{t,h+1}(s_{h+1}) - \left[\hat{\mathbb{P}}_{t,h}\hat{V}_{t,h+1}\right](s_h,a_h)\right]$$

$$+ \sum_{h\in[H]}\mathbb{E}_{d_0,\pi_t^1,\mathbb{P}^*}\left[\left[\hat{\mathbb{P}}_{t,h}\hat{V}_{t,h+1}\right](s_h,a_h) - \hat{V}_{t,h+1}(s_{h+1})\right]$$

$$= \sum_{h\in[H]}\mathbb{E}_{d_0,\pi_t^2,\mathbb{P}^*}\left[\hat{V}_{t,h+1}(s_{h+1}) - \left[\hat{\mathbb{P}}_{t,h}\hat{V}_{t,h+1}\right](s_h,a_h)\right]$$

$$+ \sum_{h\in[H]}\mathbb{E}_{d_0,\pi_t^2,\widetilde{\mathbb{P}}_t}\left[\hat{V}_{t,h+1}(s_{h+1}) - \left[\hat{\mathbb{P}}_{t,h}\hat{V}_{t,h+1}\right](s_h,a_h)\right]$$

$$- \sum_{h\in[H]}\mathbb{E}_{d_0,\pi_t^2,\mathbb{P}^*}\left[\hat{V}_{t,h+1}(s_{h+1}) - \left[\hat{\mathbb{P}}_{t,h}\hat{V}_{t,h+1}\right](s_h,a_h)\right]$$

$$+ \sum_{h\in[H]}\mathbb{E}_{d_0,\pi_t^1,\mathbb{P}^*}\left[\left[\hat{\mathbb{P}}_{t,h}\hat{V}_{t,h+1}\right](s_h,a_h) - \hat{V}_{t,h+1}(s_{h+1})\right]$$

$$\leq 2B \cdot \sum_{j\in\{1,2\}}\sum_{h\in[H]}\mathbb{E}_{d_0,\pi_t^j,\mathbb{P}^*}\left[\text{TV}(\hat{\mathbb{P}}_{t,h}(\cdot|s_h,a_h)),\mathbb{P}_h^*(\cdot|s_h,a_h)\right]$$

$$+ 2BH \cdot \text{TV}(\{d_0,\pi_t^2,\widetilde{\mathbb{P}}_t\},\{d_0,\pi_t^2,\mathbb{P}^*\})$$

$$\leq 2B \cdot \underbrace{\sum_{j\in\{1,2\}}\sum_{h\in[H]}\mathbb{E}_{d_0,\pi_t^j,\mathbb{P}^*}\left[\text{TV}(\hat{\mathbb{P}}_{t,h}(\cdot|s_h,a_h)),\mathbb{P}_h^*(\cdot|s_h,a_h)\right]}_{\text{term (D)}_t}$$

$$+ 2BH \cdot \underbrace{\sum_{j\in\{1,2\}}\sum_{h\in[H]}\mathbb{E}_{d_0,\pi_t^j,\mathbb{P}^*}\left[\text{TV}(\widetilde{\mathbb{P}}_{t,h}(\cdot|s_h,a_h)),\mathbb{P}_h^*(\cdot|s_h,a_h)\right]}_{\text{term (C)}_t}.$$

In the above derivations, we have repeatedly used similar relationships as follows:

$$\mathrm{TV}(\{d_0, \pi_t^2, \widetilde{\mathbb{P}}_t\}, \{d_0, \pi_t^2, \mathbb{P}^*\}) \leq \sum_{h \in [H]} \mathbb{E}_{d_0, \pi_t^2, \mathbb{P}^*} \left[ \mathrm{TV}\left( \widetilde{\mathbb{P}}_{t,h}(\cdot | s_h, a_h), \mathbb{P}_h^*(\cdot | s_h, a_h) \right) \right],$$

which can be derived as

$$\mathrm{TV}(\{d_0, \pi_t^2, \widetilde{\mathbb{P}}_t\}, \{d_0, \pi_t^2, \mathbb{P}^*\}) \leq \sum_{h \in [H]} \mathrm{TV}\left( \{d_0, \pi_t^2, \mathbb{P}_{1:h-1}^*, \widetilde{\mathbb{P}}_{t,h:H}\}, \{d_0, \pi_t^2, \mathbb{P}_{1:h}^*, \widetilde{\mathbb{P}}_{t,h+1:H}\} \right)$$

$$= \sum_{h \in [H]} \mathbb{E}_{d_0, \pi_t^2, \mathbb{P}^*} \left[ \mathrm{TV}\left( \widetilde{\mathbb{P}}_{t,h}(\cdot | s_h, a_h), \mathbb{P}_h^*(\cdot | s_h, a_h) \} \right) \right].$$

Then, we can obtain that

$$\sum_{t \in [T]} J(\pi^*) - J(\hat{\pi}_t^1) \leq \sum_{t \in [T]} \text{term (A)}_t + \sum_{t \in [T]} \text{term (B)}_t$$

$$+ (4B + 2BH) \sum_{t \in [T]} \text{term (C)}_t + 2B \sum_{t \in [T]} \text{term (D)}_t.$$

Then, we control the sum of each individual term in the following. First, for term $(A)_t$, with probability at least $1 - \delta$, we have that

$$\sum_{t \in [T]} \text{term (A)}_t$$

$$= \sum_{t \in [T]} \mathbb{E}_{d_0, \pi_t^2, \mathbb{P}^*} \left[ \widetilde{u}_t(s_H, a_H) \right] - \mathbb{E}_{d_0, \pi_t^1, \mathbb{P}^*} \left[ \widetilde{u}_t(s_H, a_H) \right] - \left( \mathbb{E}_{d_0, \pi_t^2, \mathbb{P}^*} \left[ u^*(s_H, a_H) \right] - \mathbb{E}_{d_0, \pi_t^1, \mathbb{P}^*} \left[ u^*(s_H, a_H) \right] \right)$$

$$\leq \sum_{t \in [T]} \widetilde{u}_t(s_{t,H}^2, a_{t,H}^2) - \widetilde{u}_t(s_{t,H}^1, a_{t,H}^1) - \left( u^*(s_{t,H}^2, a_{t,H}^2) - u^*(s_{t,H}^1, a_{t,H}^1) \right) + O(B\sqrt{T \log(1/\delta)})$$

$$\leq \sqrt{d_{\mathcal{U}} \sum_{t=2}^{T} \left( 1 + \sum_{i=1}^{t-1} \left( \widetilde{u}_t(s_{i,H}^2, a_{i,H}^2) - \widetilde{u}_t(s_{i,H}^1, a_{i,H}^1) - \left( u^*(s_{i,H}^2, a_{i,H}^2) - u^*(s_{i,H}^1, a_{i,H}^1) \right) \right)^2 \right)}$$

$$+ O(B\sqrt{T \log(1/\delta)})$$

$$\leq \sqrt{d_{\mathcal{U}} \sum_{t=2}^{T} \left( 1 + \kappa^{-2} \sum_{i=1}^{t-1} \left( \sigma\left( \widetilde{u}_t(s_{i,H}^2, a_{i,H}^2) - \widetilde{u}_t(s_{i,H}^1, a_{i,H}^1) \right) - \sigma\left( u^*(s_{i,H}^2, a_{i,H}^2) - u^*(s_{i,H}^1, a_{i,H}^1) \right) \right)^2 \right)}$$

$$+ O(B\sqrt{T \log(1/\delta)})$$

$$\lesssim \kappa^{-1} B \sqrt{d_{\mathcal{U}} T \log(|\mathcal{U}|T/\delta)},$$

where the first inequality is from the Hoeffding inequality, the second inequality uses the Eluder coefficient $d_{\mathcal{U}} := \mathrm{EC}(1, \mathcal{U} - \mathcal{U}, T)$ from Definition 3, the third inequality leverages the mean value theorem with $\kappa := 1/(2 + \exp(-B) + \exp(B))$ representing the minimum derivative of $\sigma(\cdot)$ in the regime of $[0, B]$, and the last inequality incorporates Lemma 2. A similar result can be obtained for term $(B)_t$.

For term $(C)_t$, we have that

$$\sum_{t \in [T]} \text{term (C)}_t = \sum_{j \in \{1,2\}} \sum_{t \in [T]} \sum_{h \in [H]} \mathbb{E}_{d_0, \pi_t^j, \mathbb{P}^*} \left[ \mathrm{TV}\left( \widetilde{\mathbb{P}}_{t,h}(\cdot | s_h, a_h), \mathbb{P}_h^*(\cdot | s_h, a_h) \right) \right]$$

$$= \sum_{j \in \{1,2\}} \sum_{t \in [T]} \sum_{h \in [H]} \mathrm{TV}\left( \{d_0, \pi_t^j, [\mathbb{P}_{1:h-1}^*, \widetilde{\mathbb{P}}_{t,h}, \mathbb{P}_{h+1:H}^*]\}, \{d_0, \pi_t^j, \mathbb{P}_{1:H}^*\} \right)$$

$$\leq 2H \cdot \xi(d_{\mathcal{P}}, T, c_2 \log(|\mathcal{P}|HT/\delta)),$$

where the last step is from the generalized Eluder-type condition in Definition 4 and Lemma 2. A similar result can be obtained for term $(D)_t$.

Finally, we obtain that

$$\mathrm{Reg}(T) \lesssim \kappa^{-1} B \sqrt{d_{\mathcal{U}} T \log(|\mathcal{U}|T/\delta)} + B^2 H \xi(d_{\mathcal{P}}, T, c_2 \log(|\mathcal{P}|HT/\delta)$$

$$- \eta \cdot \sum_{t \in [T]} \sum_{h \in [H]} \mathbb{E}_{d_0, \pi^*, \mathbb{P}^*} \left[ D_{\mathrm{KL}}(\pi_h^*(\cdot|s_h), \pi_{t,h}^1(\cdot|s_h)) \right],$$

which concludes the proof. □

## E    TECHNICAL LEMMAS

**Lemma 3** (Solution of KL-regularized Optimization (Proposition 7.16 and Theorem 15.3 of Zhang (2023)))**.** *Given a loss functional with respect to $p(\cdot|x)$, written as*

$$\mathbb{E}_{w \sim p(\cdot)} \left[ - U(w) + \eta D_{\mathrm{KL}}\big(p(\cdot), p_0(\cdot)\big) \right]$$

$$= \eta D_{\mathrm{KL}}\Big(p(\cdot), p_0(\cdot) \exp\Big(\frac{1}{\eta} U(\cdot)\Big)\Big) - \eta \cdot \log \underbrace{\mathbb{E}_{w \sim p_0(\cdot)} \exp\Big(\frac{1}{\eta} U(w)\Big)}_{C_r},$$

*where the minimizer of the loss functional is $p^*(w) = \frac{1}{C_r} p_0(w) \exp\Big(\frac{1}{\eta} U(w)\Big)$, also known as Gibbs distribution.*

**Definition 3** (Eluder Coefficient, Definition 17.17 in Zhang (2023))**.** *Given a function class $\mathcal{F}$, its Eluder coefficient $EC(\lambda, \mathcal{F}, T)$ is defined as the smallest number $d$ so that for any sequence $\{x_t : t \in [T]\}$ and $\{f_t : t \in [T]\} \in \mathcal{F}$,*

$$\sum_{t=2}^{T} |f_t(x_t) - f^*(x_t)| \le \sqrt{d \sum_{t=2}^{T} \left( \lambda + \sum_{i=1}^{t-1} (f_t(x_i) - f^*(x_i))^2 \right)}.$$

**Definition 4** (Generalized Eluder-type Condition, Condition 3.1 in Liu et al. (2023a))**.** *There exists a real number $d_{\mathcal{P}} \in \mathbb{R}^+$ and a function $\xi$ such that for any $(T, \Delta) \in \mathbb{N} \times \mathbb{R}^+$, transitions $\{\mathbb{P}'_t : t \in [T]\}$ and policies $\{\pi_t : t \in [T]\}$, we have*

$$\forall t \in [T], \quad \sum_{i < t} \mathrm{TV}(\{d_0, \mathbb{P}'_i, \pi_i\}, \{d_0, \mathbb{P}, \pi_i\})^2 \le \Delta$$

$$\Rightarrow \sum_{t \in [T]} \mathrm{TV}(\{d_0, \mathbb{P}'_t, \pi_t\}, \{d_0, \mathbb{P}, \pi_t\}) \le \xi(d_{\mathcal{P}}, T, \Delta).$$

