# OpenReview forum: "Building Math Agents with Multi-Turn Iterative Preference Learning"
_ICLR.cc/2025/Conference — ICLR 2025 Poster_

### Official Review · Reviewer_h8KG · 2024-10-19

**Soundness:** 2
**Presentation:** 3
**Contribution:** 3
**Rating:** 6
**Confidence:** 4

**Summary:**

This paper proposes a new method to improve the mathematical reasoning abilities of large language models. Existing direct preference learning algorithms are mainly applied to single-turn dialogue tasks and fail to address the complexities of multi-turn reasoning and external tool integration. To tackle this, the paper introduces a multi-turn direct preference learning framework and specifically implements two algorithms (M-DPO and M-KTO). Experiments were conducted by training the model on the GSM8K and MATH datasets, and the results demonstrate that this new framework significantly enhances the model’s performance on multiple benchmark tasks.

**Strengths:**

This paper improves the performance of large language models (LLMs) in multi-turn complex reasoning tasks. The study effectively validates the proposed approach using two datasets.

**Weaknesses:**

Although the proposed method shows outstanding performance, it is relatively focused on specific task types. Currently, the experiments are limited to the GSM8K and MATH datasets, and it would be better to include more datasets in the testing. Additionally, the framework's design is relatively complex, involving multi-turn preference learning and trajectory-level optimization, which may require high computational resources in practical applications. A comparative experiment to demonstrate the time cost and computational consumption compared to previous methods would be helpful here.

**Questions:**

1.How is the definition and selection of preference signals specifically designed in this approach? Could different preference signals significantly impact the final optimization results?
2.Is it possible to further test the method in other types of reasoning tasks or application domains to verify its generalizability?
3.In terms of external tool integration, is the current method the optimal solution?

---

> ### Author Response · Authors · 2024-11-20
> **Thanks for the efforts in reviewing our paper and the constructive feedback!**
>
> Thanks for the efforts in reviewing our paper and the constructive feedback!
>
> **Q1: Is it possible to further test the method in other types of reasoning tasks or application domains to verify its generalizability?**
>
> Thanks for the feedback and we agree that the results with more complex tasks would be more interesting. In the current version, we choose the tool-integrated reasoning task for math problem mainly because it is a well-established task in the literature and is suitable for our purpose of illustrating the benefits of multi-turn preference learning.
>
> While we focus on the math problem, the mathematical formulation itself is general enough to capture any multi-turn problems with the presence of external observations. In particular, we would like to share some preliminary result we have for the follow-up study. In Equation (7), we directly discard term (C) because for the tool-integrated reasoning, the external environment is almost deterministic given the history. However,  our theoretical framework itself is more general than the presented algorithm. The main intuition is that according to Equation (4), the optimal value itself can be characterized by the distribution of the reference model, making the value estimation possible. We do agree that some more complex problems are interesting, especially for the case with stochastic external environments or against dynamic opponents. We hope these clarifications and discussions can convey enough information in terms of the generality of the framework established in this paper.
>
> **Q2: In terms of external tool integration, is the current method the optimal solution?**
>
> Thanks for the question! The optimality condition described in Equation (3) and (4) are derived from the KL-regularized target, without other assumptions. However, building on these optimality conditions, the algorithmic designs can be different.
>
> For instance, one can try to do the value estimation based on Equation (4) and use the deep RL methods to improve the policy. Unfortunately, in the context of LLM, the deep RL methods are extremely challenging to tune to their best performance. Indeed, almost all the decent open-source models are trained by the direct preference learning algorithms such as DPO, IPO, KTO, or Slic-HF. Therefore, there is a trade-off between the theoretical upper bound of the performance and practical applicability. This is the main reason why we choose to do the algorithm designs under the framework of direct preference algorithm.
>
> The main drawback of the current method is that, we impose an additional Bradley-Terry assumption, which can be imperfect due to its limited capacity. In recognition of this, one possible choice is to replace the BT assumption with the general pairwise preference model with larger capacity. This process can be a routine practice based on the previous works in the single-turn scenario [1, 2].
>
> **Q3: time cost and computational consumption**
>
> Thanks for the suggestion! For the multi-turn direct preference learning framework, the previous methods mainly do the regular DPO training that computes the loss on **all** the tokens including the external messages. Compared to them, our methods will not introduce much computational overhead. The overall time used in our experiments is relatively the same. We will include a discussion on this in the next revision. Thanks for the feedback.
>
>
>
> **Q4: How is the definition and selection of preference signals specifically designed in this approach? Could different preference signals significantly impact the final optimization results?**
>
> We use the final result checking to check the correctness of the final answer in the experiment and construct the pairwise comparison by randomly sampling one correct and one incorrect trajectory. One may further use step-wise reward signal in the data ranking stage and also leverage the LLM-as-judge to go through all the intermediate steps. We believe that this can be beneficial because in our experiments, we do observe a small portion of samples are with correct final answer but incorrect reasoning steps. However, these require the PRM construction (which is the Q estimation, an interesting observation from our theoretical framework), so we leave it for future study.
>
> [1] Nash Learning from Human Feedback
>
> [2] Direct nash optimization: Teaching language models to self-improve with general preferences

---

### Official Review · Reviewer_erZb · 2024-10-28

**Soundness:** 4
**Presentation:** 4
**Contribution:** 3
**Rating:** 8
**Confidence:** 4

**Summary:**

This paper introduces a multi-turn direct preference learning framework, tailored for this context, that leverages feedback from code interpreters and optimizes trajectory-level preferences. This framework includes multi-turn DPO and multi-turn KTO as specific implementations. The effectiveness of our framework is validated through training of various language models using an augmented
prompt set from the GSM8K and MATH datasets.

**Strengths:**

1. The integration of external tool augmented methods and the use of multi-turn preference learning is both interesting and reasonable, effectively addressing the shortcomings of natural language in mathematical reasoning. The exploration of tool-augmented methods in the mathematical domain is well-motivated.

2. The methods presented in the paper are strongly supported by theoretical foundations, which enhances their credibility, and the overall writing is clear and easy to understand.

**Weaknesses:**

1. WizardMath, being a state-of-the-art math model from a year ago, seems somewhat outdated as a baseline for performance. I hope the authors can include results from recent stronger models in mathematics that incorporate MULTI-TURN ITERATIVE PREFERENCE LEARNING, such as Qwen 2.5 Math and DeepSeek Math.

2. I would indeed like to see results of this method in the context of a process-based reward model (PRM) and combined with MCTS sampling strategies (even preliminary results would be helpful). Although the authors mention this limitation in the limitations section and in line 297, relying solely on ORM for reasoning verification is not sufficiently detailed.

**Questions:**

Please see the weaknesses section. I also hope to see further details on the ORM, such as the base model and training method.

Overall, the paper does not have any obvious weaknesses, and I personally find it quite impressive. I believe it meets the acceptance criteria for ICLR. If you address my concerns, I would consider raising the score.

**Missing reference:**

[1] DotaMath: Decomposition of Thought with Code Assistance and Self-correction for Mathematical Reasoning

---

> ### Author Response · Authors · 2024-11-20
> **Thanks for the efforts in reviewing our paper!**
>
> Thanks for the efforts in reviewing our paper and the constructive feedback! Our responses are as follows.
>
> **Q1: WizardMath, being a state-of-the-art math model from a year ago, seems somewhat outdated as a baseline for performance. I hope the authors can include results from recent stronger models in mathematics that incorporate MULTI-TURN ITERATIVE PREFERENCE LEARNING, such as Qwen 2.5 Math and DeepSeek Math.**
>
> Thanks for the suggestion! We would like to clarify that the WizardMath mainly serves to motivate the use of the tool-integrated reasoning. We realize that we should have omitted the results from the main table since the advantage of the tool-integrated reasoning is relatively well-accepted in the community (e.g. [1]). Instead, we can include a short paragraph to discuss the advantage of tool use with appropriate reference.
>
> The comparison with the state-of-the-art Qwen and Deepseek models can be more tricky since the superior performances of these models rely on industry-level resources. For instance, Deepseek Math was trained from the pre-training stage to boost its mathematical reasoning ability and Qwen math uses a diverse set of 824K mathematical problems and heavy rejection sampling. This is the main reason why we focus on the *relative improvement* across many different popular base models to verify the effectiveness of our methods. **We will include a short paragraph to include the results of these models and add appropriate discussion accordingly in the next revision.**
>
> We also want to clarify that the main goal is paper is to develop a machine learning framework for the multi-turn problem, but rather train a state-of-the-art model (which is beyond the resource (16GPUs) we can have for a research project).
>
> **Q2: I would indeed like to see results of this method in the context of a process-based reward model (PRM) and combined with MCTS sampling strategies (even preliminary results would be helpful). Although the authors mention this limitation in the limitations section and in line 297, relying solely on ORM for reasoning verification is not sufficiently detailed.**
>
> Thanks for the suggestion! We would like to first clarify that the experiments actually use the **gold final answer checking** instead of a trained ORM. In other words, the trajectory receives a reward of 1 if the result is correct and 0 otherwise.
>
> In terms of PRM, we do observe an interesting connection between the PRM and our theoretical framework. The takeaway message is that the popular automatic annotation method [2] for PRM training is equivalent to the Q learning in our theoretical framework. The main intuition is that, according to Equation (4), the optimal V value can be characterized by the data collected by the reference model. Therefore, we can use the Monte-Carlo estimation to estimate the Q value and do Q learning. Specifically, starting from a fixed step h, and state-action pair $(s_h, a_h)$, we can use the current model to sample M=8 completions, and use these completions to estimate the Q value.
>
> [2] studies the case of non-regularized framework, where they are essentially estimating the $Q^\pi$. We are indeed aware of some follow-up work that follows the theoretical framework of this paper to do the regularized version of PRM training. Unfortunately, the recipe to do the PRM training takes more than 30 days to collect the data with 8 x A100 and VLLM. Also, this can lead to some conflict of interests issue if we include them into the revision. We thanks the reviewers for the constructive suggestions in terms of the further PRM (Q estimation) design, which can also be leveraged into the algorithm design in order to extend the algorithm to handle the general scenarios.
>
> We hope these clarifications and discussions can convey enough information for the follow-up Q estimation and PRM-related study.
>
> [1] Program of thoughts prompting: Disentangling computation from reasoning for numerical reasoning tasks.
>
> [2] Math-Shepherd: Verify and Reinforce LLMs Step-by-step without Human Annotations.
>
> **Missing reference**
>
> Thanks for bringing this to our attention! We will include it in the next revision.

---

> > ### Comment · Reviewer_erZb · 2024-11-21
> > **Thanks for reviewer's response**
> >
> > I think the author's response is reasonable and addresses most of my concerns.
> > I will improve my score and hope to add the experiments promised in the rebuttal to the latest version.

---

### Official Review · Reviewer_4eni · 2024-11-01

**Soundness:** 3
**Presentation:** 4
**Contribution:** 3
**Rating:** 8
**Confidence:** 4

**Summary:**

This paper addresses the optimization of multi-turn MDP planning under conditions involving interactions with external tool integration. Through rigorous theoretical derivations, it models the optimal conditions for multi-turn MDP planning and demonstrates that when
the external randomness is low, a reward form using DPO can feasibly compute the utility function in multi-turn MDP planning. Furthermore, the authors propose multi-turn DPO and multi-turn KTO algorithms tailored for optimizing multi-turn MDP planning and validate the effectiveness of combining multi-turn MDP planning optimization with an online iterative approach, showing that performance loss relative to the optimal policy is sublinear, thus highlighting the statistical efficiency of the proposed algorithm. Extensive experiments conducted on various model series across GSM8K and MATH datasets affirm the efficacy of these methods in multi-turn MDP planning.

**Strengths:**

1. This paper proposes a solution to the language model’s multi-turn planning problem involving external environment interaction, offering well-grounded theoretical proofs and a straightforward formulation.

2. The proposed algorithms are empirically validated to be effective in addressing mathematical problem-solving tasks with external tool integration.

**Weaknesses:**

1. The utility function form in Equation (7) needs a value network and is computable only under the form of DPO rewards and deterministic external conditions, which limits the practical applicability of this theoretical framework. Currently, the theoretic framework cannot utilize methods like PPO to address multi-turn MDP planning problems with external interactions.

2. The experimental section’s baseline explanation for single-turn iterative DPO lacks clarity regarding the KL regularization strategy, especially in terms of whether a fixed reference is employed, which might contribute to the poorer performance observed in single-turn iterative DPO.

**Questions:**

None.

---

> ### Author Response · Authors · 2024-11-20
> **Thanks for the efforts in reviewing our paper!**
>
> Thanks for the efforts in reviewing our paper and  your recognition of our work!
>
> Our responses are as follows.
>
> **Q1: The utility function form in Equation (7) needs a value network and is computable only under the form of DPO rewards and deterministic external conditions, which limits the practical applicability of this theoretical framework. Currently, the theoretic framework cannot utilize methods like PPO to address multi-turn MDP planning problems with external interactions.**
>
> Thanks for the feedback! Since the focus of this paper is the tool-integrated reasoning, we only focus on the case where the term C in Equation (7) is zero. We would like to clarify that the theoretical framework itself is more general than the presented algorithm. The main intuition is that according to Equation (4), the optimal value itself can be characterized by the distribution of the reference model, making the value estimation possible.
>
> Meanwhile, even for the PPO training, we can have some very interesting results from our theoretical framework. For instance, one can recognize that the automatic annotation method for PRM training essentially is (the non-regularized) Q value estimation via Monte-Carlo estimator. See the expression of V value in Equation (4) for an example. Therefore, when we deploying the PRM in PPO training, one is supposed to use the difference between the two steps as the reward signal. This was independently observed in the empirical community after the submission deadline of ICLR 2025 [2].
>
> We will integrate some discussions on the theoretical framework in the next revision and hope that the results can motivate more research in this direction!
>
> [1] Math-Shepherd: Verify and Reinforce LLMs Step-by-step without Human Annotations,
>
> [2] On Designing Effective RL Reward at Training Time for LLM Reasoning
>
> **Q2: The experimental section’s baseline explanation for single-turn iterative DPO lacks clarity regarding the KL regularization strategy, especially in terms of whether a fixed reference is employed, which might contribute to the poorer performance observed in single-turn iterative DPO.**
>
> Thank for the suggestion! We also do the ablation in terms of the reference model choice (fixed one or updated reference), as well as the KL regularization level (0.01, 0.1, 0.5), and the results are consistent with our main multi-turn experiment: updated reference + 0.1 KL coefficient offers the best test accuracy. We will definitely include these details in the next revision.

---

### Official Review · Reviewer_Roam · 2024-11-02

**Soundness:** 3
**Presentation:** 3
**Contribution:** 3
**Rating:** 6
**Confidence:** 3

**Summary:**

The paper investigates enhancing mathematical reasoning in large language models (LLMs) through a multi-turn preference learning approach with tool integration. The authors propose a multi-turn direct preference optimization framework that accommodates complex, multi-turn reasoning and external tool integration, crucial for mathematical tasks. By employing two specific implementations—multi-turn Direct Preference Optimization (DPO) and multi-turn Kernelized Trajectory Optimization (KTO)—the framework improves model accuracy on math benchmarks, GSM8K and MATH. Empirical results with multiple base models show notable gains, particularly for models fine-tuned with iterative multi-turn training, outperforming standard SFT and single-turn baselines. The study also highlights the importance of balancing model exploration and accuracy through KL regularization and mixture sampling to maintain performance in iterative training.

**Strengths:**

a.	The paper proposed an iterative multi-turn DPO and KTO training framework, which effectively improves the models ability on multiturn reasoning for math problems.
b.	The experiments are solid, and they conduct experiments with multiple base models.

**Weaknesses:**

a.	Overall, the motivation makes sense, as for some complex math problems that require tool integration, human inputs might be needed. However, only GSM8K and MATH have been used as the benchmarks in this paper, which are relatively easy problems, and may not necessarily need human input as multi-turn conversation. It is more convincing if you can conduct experiments on more complex tasks, like formulating an optimization problem and solving it from a task description.
b.	The presentation of the results is not clear enough. For example, table 1 is too long, and different models use different settings, e.g. Wizardmath, codellama and Gemma-1.1-it-7B have different settings from other models. It is more readable if the table is splitted, and each sub table has its own conclusion or findings.
c.	Typos, e.g. line 270 DKO, should be KTO.

**Questions:**

For the example shown in figure 1, the human inputs (o1 and o2) do not provide additional information, why multi-turn conversation is needed in this case?

---

> ### Author Response · Authors · 2024-11-20
> **Thanks for the efforts in reviewing our paper!**
>
> Thanks for the efforts in reviewing our paper and also thanks for the constructive feedback on the presentation and writing! We will take your suggestions, fix the typo and revise the presentations accordingly in the next revision!
>
> Our responses to other questions are as follows.
>
> **Q1: For the example shown in figure 1, the human inputs (o1 and o2) do not provide additional information, why multi-turn conversation is needed in this case? …. MATH and GSM8K may not necessarily need human input as multi-turn conversation…**
>
> Thanks for the question!
>
> We would like to first clarify that o1 and o2 (the user turn) are not human input but the code execution results from the external python interpreter (i.e., tool outputs). The tool-integrated reasoning (with python interpreter) is a well-established setting and proves to be able to largely improve the reasoning ability of the LLM in the literature [1, 2]. While the previous works mainly focus on synthetic data generation and SFT, we focus on the preference learning approach in this project. Furthermore, we would also like to clarify that while we focus on the tool-integrated reasoning, the mathematical formulation and theoretical result apply to any multi-round problems with interactions with the external environment.
>
> In terms of the example in Figure 1, we use the current example because it is relatively short and can illustrate the main format of the whole problem. But your feedback makes us realize that we could have improved the presentation by a more interesting example. In particular, in our experiments , we find that **the model can fix the error in the previous code based on the error messages returned by the environment**, where this is a more interesting examples to motivate the multi-turn conversation. We will add this example in the next revision of paper.
>
> [{'content': 'If $z = 2y$ and $y = \\displaystyle\\frac{1}{3x+1}$, what is the value of $x$ when $z = 2$?', 'role': 'user'},
>
> {'content': "We can solve this problem using algebraic operations.\nUsing sympy, we can simplify the process:\n```python\nfrom sympy import symbols, Eq, solve\n\n# Define the variables\nx = symbols('x')\ny = symbols('y')\nz = symbols('z')\n\n# Given equations:\neq1 = Eq(z, 2 * y)\neq2 = Eq(y, 1 / (3 * x + 1))\n\n# Solve the equations:\nsolution = solve([eq1, eq2], [x, y])\n\n# Extract the result for x\nx_result = solution[x]\n\n# Print the result\nprint(x_result)\n```", 'role': 'assistant'},
>
> {'content': '```output\nExecution error: list indices must be integers or slices, not Symbol\n```', 'role': 'user'},
>
> {'content': "Unfortunately, this code gives an error.\nSo we need to use a different approach.\nIf $z = 2y$ and $y = 1/(3x + 1)$, then $y$ and $z$ are related by $z = 2 / (3x + 1)$.\nSince $z = 2$, we can substitute this value into the equation and solve for $x$:\n```python\n\n# Define the variables\nx = symbols('x')\nz = symbols('z')\n\n# Given equation:\neq = Eq(2, 2 / (3 * x + 1))\n\n# Solve the equation:\nsolution = solve(eq, x)\n\n# Print the result\nprint(solution)\n```", 'role': 'assistant'},
>
> {'content': '```output\n[0]\n```', 'role': 'user'}, {'content': 'Thus, the value of $x$ when $z = 2$ is $\\boxed{0}$.', 'role': 'assistant'}]
>
> **Q2: It is more convincing if you can conduct experiments on more complex tasks, like formulating an optimization problem and solving it from a task description.**
>
> Thanks for the suggestion and we agree that the results with more complex tasks would be more interesting.
>
> We choose the tool-integrated reasoning task for math reasoning mainly because it is a well-established task in the literature and is suitable for our purpose of illustrating the benefits of multi-turn preference learning. We notice that with the example presented in the previous response, the external observation for the math problem can be also be very informative that requires the model’s ability to fix its error based on the error message.
>
> We would also like to clarify that while we focus on the math problem, our mathematical formulation itself is general enough to cover any multi-turn problems with the presence of external observations. We do agree that some more complex problems are interesting, especially for the case with stochastic external environments or against dynamic opponents as we discuss in the conclusion and future research direction. We are aware of some follow-up works focusing on this direction, where they combine the framework with value estimation, which are also described in our conclusion part. The main intuition is that, according to Equation (4) of our paper, in the KL-regularized framework, the optimal value can be estimated by the reference model. We hope our work can motivate more involved algorithm design in the future work.
>
> [1] Program of thoughts prompting: Disentangling computation from reasoning for numerical reasoning tasks.
>
> [2] ToRA: A Tool-Integrated Reasoning Agent

---

### Comment · Area_Chair_apXt · 2024-11-25
**Reviewer Response**

Dear Reviewers,

The rebuttal period is almost over, and the paper has received both positive and negative feedback. The authors have worked hard to address all your concerns. Could you take a moment to review their responses and let us know if you have any unresolved issues?

Best, AC

---

### Meta-Review · Area_Chair_apXt · 2024-12-23

**Metareview:**

The paper introduces a multi-turn direct preference learning framework designed to enhance large language models’ (LLMs) mathematical reasoning by addressing limitations of single-turn preference learning. It focuses on trajectory-level preferences and integrates two specific algorithms: multi-turn Direct Preference Optimization and Kernelized Trajectory Optimization. Experiments on GSM8K and MATH benchmarks demonstrate notable performance improvements across various model sizes.

Reviewers appreciated the paper’s theoretical framework, practical application, and integration of external tools for tool-based reasoning tasks. The paper validated the method with solid experimental setup, including ablation studies and detailed analyses. While no major weaknesses were identified, reviewers suggested that the benchmarks used may not fully demonstrate generalizability to complex scenarios, and clarity on computational costs and result presentation could be improved. The rebuttal has addressed key concerns, and reviewers agreed the work provides significant theoretical and practical contributions. Incorporating the feedback in revisions would further strengthen this already compelling paper. I recommend it for acceptance.

**Additional Comments On Reviewer Discussion:**

During the rebuttal, the authors clarified detailed responses to concerns about scalability, task generalizability, and computational efficiency, promising to incorporate missing discussions and improve presentation in the final revision. These efforts were well-received by reviewers. Some noted that addressing more complex problems and incorporating cutting-edge baselines would further strengthen the work’s impact.

---

### Decision · Program_Chairs · 2025-01-22

Accept (Poster)